# MAPLE: Multi-scale Attribute-enhanced Prompt Learning for Few-shot Whole Slide Image Classification

Junjie Zhou[1,2], Wei Shao[1,2],[*] Yagao Yue[1,2], Wei Mu[3], Peng Wan[1,2], Qi Zhu[1,2], Daoqiang Zhang[1,2]

[1]The College of Artificial Intelligence, Nanjing University of Aeronautics and Astronautics
[2]The Key Laboratory of Brain-Machine Intelligence Technology, Ministry of Education
[3]The School of Engneering Medicine, Beihang University
junjiezhou@nuaa.edu.cn, shaowei20022005@nuaa.edu.cn, dqzhang@nuaa.edu.cn

## Abstract

Prompt learning has emerged as a promising paradigm for adapting pre-trained vision-language models (VLMs) to few-shot whole slide image (WSI) classification by aligning visual features with textual representations, thereby reducing annotation cost and enhancing model generalization. Nevertheless, existing methods typically rely on slide-level prompts and fail to capture the subtype-specific phenotypic variations of histological entities (*e.g.,* nuclei, glands) that are critical for cancer diagnosis. To address this gap, we propose Multi-scale Attribute-enhanced Prompt Learning (**MAPLE**), a hierarchical framework for few-shot WSI classification that jointly integrates multi-scale visual semantics and performs prediction at both the entity and slide levels. Specifically, we first leverage large language models (LLMs) to generate entity-level prompts that can help identify multi-scale histological entities and their phenotypic attributes, as well as slide-level prompts to capture global visual descriptions. Then, an entity-guided cross-attention module is proposed to generate entity-level features, followed by aligning with their corresponding subtype-specific attributes for fine-grained entity-level prediction. To enrich entity representations, we further develop a cross-scale entity graph learning module that can update these representations by capturing their semantic correlations within and across scales. The refined representations are then aggregated into a slide-level representation and aligned with the corresponding prompts for slide-level prediction. Finally, we combine both entity-level and slide-level outputs to produce the final prediction results. Results on three cancer cohorts confirm the effectiveness of our approach in addressing few-shot pathology diagnosis tasks. Codes will be available at `https://github.com/JJ-ZHOU-Code/MAPLE`.

## 1 Introduction

Whole slide images (WSIs) have become the clinical gold standard for cancer diagnosis, offering gigapixel-resolution views of tissue architecture and cellular morphology [5, 6]. However, their huge size (*e.g.,* 150,000 × 150,000 pixels) and hierarchical structure render dense annotation of individual patches impractical. To overcome this challenge, multiple instance learning (MIL) has emerged as an effective way for weakly supervised WSI analysis [18, 32, 23, 39, 9], where each WSI is divided into thousands of patches, encoded via a pre-trained feature extractor, and aggregated into a slide-level representation for classification [17]. Beyond weak supervision, WSI classification faces another critical challenge: the scarcity of the labeled images [26, 31]. This limitation stems from factors such

---

[*]Corresponding author

39th Conference on Neural Information Processing Systems (NeurIPS 2025).

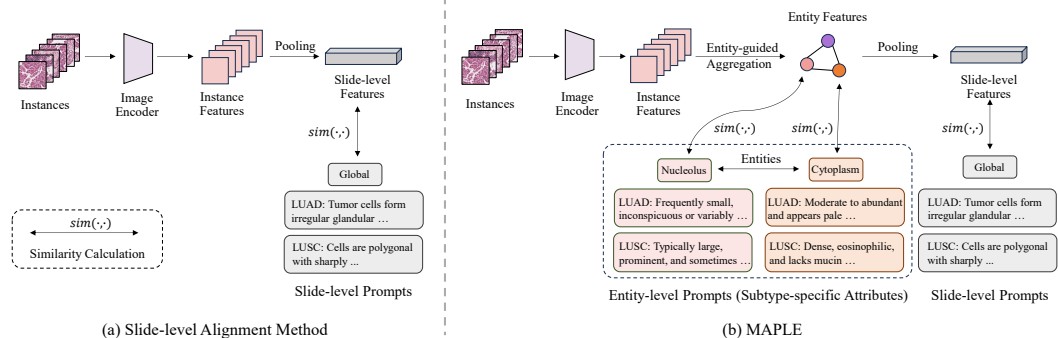

**Figure 1:** Comparison of MAPLE with existing slide-level alignment methods for the classification of lung adenocarcinoma (LUAD) and lung squamous cell carcinoma (LUSC). (a) Existing methods align slide-level features with corresponding prompts for classification. (b) Our proposed MAPLE introduces additional entity-level features and incorporates subtype-specific phenotypic attributes for more interpretable and precise alignment. For simplicity, only the single-scale data stream of MAPLE is visualized.

as privacy constraints, the difficulty of acquiring expert-labeled slides, and the low prevalence of certain cancer subtypes [12, 37, 16, 36]. Consequently, few-shot learning has become an attractive paradigm for developing robust classifiers on limited labeled data.

Recent progress in vision–language models (VLMs), such as CLIP [30], offers a promising path for few-shot learning by aligning image and text representations in a shared embedding space. Building upon VLMs, prompt learning techniques [44, 43, 25, 21, 4] adapt textual inputs using a small number of labeled examples, enabling effective transfer to new tasks without fine-tuning the vision backbone. However, applying these methods for WSI classification remains non-trivial. Unlike natural images, WSIs are extremely large and are typically divided into thousands of instances, making it difficult to construct a unified visual representation suitable for prompt alignment [29, 10]. At the same time, designing prompts that accurately reflect the complex tissue morphology and subtype-specific patterns within a WSI is also challenging. Simple prompts like "a WSI of [CLASS]" often fail to capture the localized, fine-grained attributes that are crucial for cancer diagnosis [34]. These limitations highlight a central question: *how can we bridge the gap between fine-grained instance-level visual details and semantically rich prompts for effective few-shot WSI classification?*

Recently, several methods have attempted to work on it. For instance, TOP [29] introduces instance-level phenotypic prompts to guide patch aggregation into slide-level features, while ViLa-MIL [34] leverages learnable visual prototypes to guide the fusion process of patch features and considers dual-scale visual descriptive text prompt to boost the performance. However, these approaches generally focus solely on slide-level feature alignment after the aggregation of instance-level representations, and fail to capture the subtype-specific phenotypic variations of histological entities (*e.g.,* nuclei, cytoplasm, glands) that are critical for cancer diagnosis. For instance, nucleoli are typically small and inconspicuous in lung adenocarcinoma (LUAD) but appear large and prominent in lung squamous cell carcinoma (LUSC). Furthermore, different resolution levels in WSIs naturally correspond to different scales of histological entities, where low magnification reveals tissue architecture and organization patterns, while high magnification exposes cellular details and nuclear morphology. Ignoring such fine-grained, multi-scale entity variations limits the model's ability to capture discriminative patterns and reduces interpretability for cancer diagnosis.

To this end, we propose Multi-scale Attribute-enhanced Prompt Learning (**MAPLE**), a hierarchical framework designed for few-shot WSI classification by the combination of entity-level and slide-level predictions. Different from the previous slide-level alignment methods [29, 34, 10, 11], MAPLE additionally considers that the diagnostic information among different cancer subtypes is also reflected by the phenotypic attributes of histological entities across different scales, as illustrated in Fig. 1. Specifically, we begin by leveraging large language models (LLMs) to construct two types of prompts: entity-level prompts that identify multi-scale histological entities and their distinctive phenotypic characteristics, and slide-level prompts that capture comprehensive global visual patterns. After employing language-guided instance selection strategy to identify discriminative tumor-related patches from WSIs, we subsequently introduce an entity-guided cross-attention module to extract entity-level features, which are then aligned with their respective subtype-specific attributes to enable

entity-level predictions. To enrich entity representations, we further develop a cross-scale entity graph learning module that can update these representations by capturing their semantic correlations within and across scales. The refined entity representations are aggregated to construct a slide-level representation, which is then aligned with the corresponding prompts to enable slide-level prediction. Finally, we combine both entity-level and slide-level outputs to produce the final prediction results. We conduct experiments on three cancer cohorts derived from the cancer genome atlas (TCGA), and the experimental results indicate the advantage of MAPLE on few-shot pathology diagnosis tasks.

## 2    Related Work

### 2.1    Multiple Instance Learning for WSI Classification

Due to the gigapixel size of WSIs and the infeasibility of dense patch-level annotation, MIL based methods [3, 28, 33, 13, 32] have become the prevailing approaches for WSI analysis. In the MIL framework, each WSI is represented as a bag of instances (patches), with only slide-level labels available for supervision. Early MIL approaches aggregate instance features using non-parametric max or mean pooling operations. Subsequent approaches introduce attention-based pooling mechanisms that learn to assign importance weights to instances, significantly enhancing discriminative power and classification performance [15, 32, 39, 41, 40]. Recent works have further advanced the MIL framework with more structured representations. For example, GTP [42] introduces a graph-based vision transformer that models WSIs using sparse token selection and inter-instance relations. WiKG [19] proposes a novel dynamic graph representation algorithm that conceptualizes WSIs as a form of the knowledge graph structure. While these methods demonstrate strong performance under full supervision, they typically require large annotated datasets for training, making them less suitable for scenarios with limited data availability, which is a common challenge in clinical settings. In this work, we propose a multi-scale attribute-enhanced prompt learning framework that combines MIL with vision-language models to enable effective WSI classification in few-shot scenarios, substantially reducing the annotation burden for practical clinical applications.

### 2.2    Prompt Learning for Few-shot WSI Classification

Prompt learning [44, 43, 25, 21, 4] has emerged as an efficient strategy for adapting VLMs to downstream tasks in data-scarce settings by optimizing only a small set of textual tokens rather than entire model parameters. Recent work has extended this paradigm to the WSI classification by combining prompt learning with MIL for few-shot classification [29, 34, 10, 11]. TOP [29] first introduces a two-level prompt learning strategy that incorporates linguistic priors to guide both instance- and slide-level feature aggregation. ViLa-MIL [34] extends this direction by proposing dual-scale visual prompts, enabling the fusion of features across high- and low-resolution magnifications. FOCUS [10] introduces a three-stage compression strategy that leverages both foundation models and language prompts for focused analysis of diagnostically relevant regions. MSCPT [11] employs a graph prompt tuning module to capture spatial context within WSIs. However, these methods primarily emphasize slide-level alignment and often overlook the fine-grained, subtype-specific phenotypic attributes of histological entities. Despite TOP first introduces a two-level framework, it solely focuses on slide-level alignment, treating histological instances implicitly through aggregation rather than modeling them as explicit diagnostic targets. Consequently, it fails to capture the fine-grained, subtype-specific phenotypic variations of nuclei, cytoplasm, glands, and other histological entities that pathologists rely on for differential diagnosis. Our MAPLE framework addresses this gap by introducing entity-level prompt learning that explicitly models these diagnostic cues across multiple scales, enabling more accurate and interpretable few-shot WSI classification.

## 3    Preliminaries

### 3.1    Problem Formulation

Given a dataset $\mathcal{X} = \{X_1, X_2, \ldots, X_N\}$ consisting of $N$ WSIs, each slide $X_i$ can be associated with a slide-level label $y_i \in \{1, 2, \ldots, C\}$, where $C$ is the number of diagnostic categories. Due to the extremely large size and high resolution of WSIs, it is computationally infeasible to process

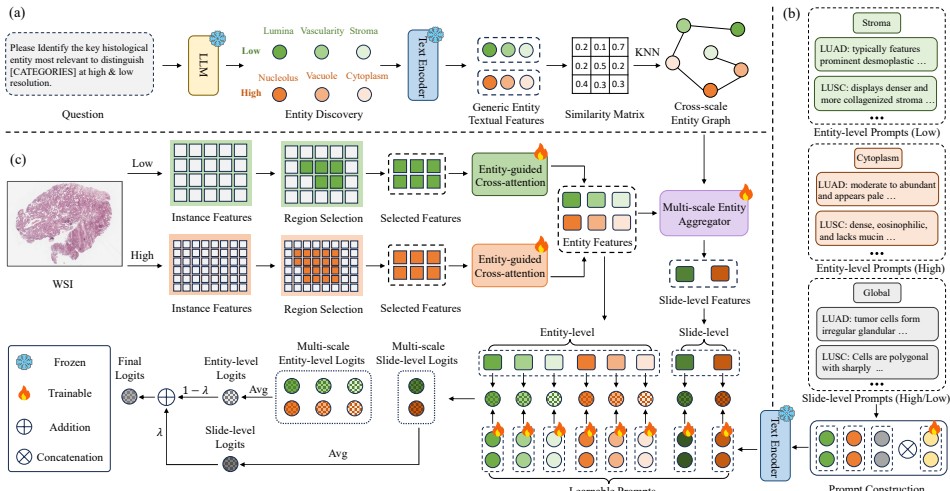

**Figure 2:** Framework of our proposed MAPLE. (a) MAPLE leverages the LLM to identify multi-scale histological entities, and then builds a cross-scale entity graph by modeling the semantic relationships wthin and across scales. (b) Both entity-level and slide-level prompts are enriched with learnable context vectors to enable effective alignment with corresponding visual features. (c) MAPLE jointly integrates multi-scale visual semantics and performs prediction at both the entity and slide levels.

entire slides directly. Instead, each WSI is divided into a set of non-overlapping $K_i$ patches $X_i = \{x_{i,1}, x_{i,2}, \ldots, x_{i,K_i}\}$, where $x_{i,j} \in \mathbb{R}^d$ denotes the feature vector of the $j$-th patch for slide $X_i$.

To address the WSI classification task, a common approach is to formulate such weakly supervised learning problem as multiple instance learning (MIL) [18, 32, 23, 9]. In the binary classification setting, MIL assumes that a bag (*i.e.,* a slide) is labeled positive if at least one of its instances is positive; otherwise, it is labeled negative [39, 17]:

$$y_i = 1 \iff \exists x_{i,j} \in X_i \text{ is positive.} \tag{1}$$

For multi-class scenarios, this formulation extends to identifying the dominant cancer subtype represented by the most discriminative patches within the slide.

In the few-shot classification setting, the problem becomes even more challenging: the objective is to learn a reliable classifier using only a limited number of labeled WSIs per class. The term "shot" refers to the number of labeled examples per class, commonly set to 1, 2, 4, 8, or 16.

### 3.2 Prompt Learning for Few Shot WSI classification

VLMs such as CLIP [30] typically consist of two parallel encoders: a vision encoder $f_v(\cdot)$ and a text encoder $f_t(\cdot)$, which are jointly trained using contrastive learning over large-scale image–text pairs. Prompt learning has emerged as a parameter-efficient strategy to adapt pre-trained VLMs to downstream tasks, such as few-shot WSI classification, without extensive fine-tuning [29, 34]. The typical pipeline for prompt learning-based few-shot WSI classification involves two key steps: instance-level feature aggregation and slide-level alignment with learnable prompts [29].

Given a WSI $X_i = \{x_{i,1}, x_{i,2}, \ldots, x_{i,K_i}\}$ partitioned into $K_i$ patches, each patch $x_{i,j}$ is first embedded using a frozen image encoder $f_v$. The instance embeddings are then aggregated via a pooling operation (*e.g.,* max, mean or attention-based) to obtain a compact slide-level representation $z_i^v = \text{Aggregate}(f_v(x_{i,j})_{j=1}^{K_i})$. Prompt learning then adapts the text encoder by introducing a set of $M$ learnable context vectors $V = \{v_1, v_2, \ldots, v_M\}$. For each class $k$, a textual prompt $t_k = \{v_1, \ldots, v_M, c_k\}$ is constructed by concatenating these context vectors with the embedding of the class name $c_k$, and passed through the text encoder to generate the class-specific textual feature $z_k^t = f_t(t_k)$. The prediction probability for class $k$ is computed based on the cosine similarity between the slide-level visual feature and the class-specific textual features:

$$P(y = k|X_i) = \frac{\exp(\text{sim}(z_i^v, z_k^t)/\tau)}{\sum_{k'=1}^{C} \exp(\text{sim}(z_i^v, z_{k'}^t)/\tau)} \tag{2}$$

where $\tau$ is a learnable temperature parameter and $\text{sim}(\cdot, \cdot)$ denotes cosine similarity. The learnable context vectors V are optimized by minimizing the standard cross-entropy loss between the prediction and the ground-truth label:

$$\mathcal{L}_{\text{CE}} = -\sum_{i=1}^{N} \log P(y = y_i \mid X_i), \tag{3}$$

where $y_i$ is the ground-truth label for slide $X_i$.

## 4 Method

In this section, we present the details of our proposed method MAPLE, Multi-scale Attribute-enhanced Prompt Learning for few-shot WSI classification. An overview of the framework is illustrated in Fig. 2. Given a WSI $X_i$, we partition it into two sets of non-overlapping patches at both high resolution *i.e.*, $X_i^h = \{x_{i,j}^h\}_{j=1}^{K_i^h}$ and low resolution *i.e.*, $X_i^l = \{x_{i,j}^l\}_{j=1}^{K_i^l}$, where $K_i^h$ and $K_i^l$ denote the number of patches at different scales. Then, each patch is embedded with a frozen vision encoder $f_v$ of the vision-language model (PLIP [14] in our implementation), resulting in the feature sets $Z_i^h = \{z_{i,j}^h\}_{j=1}^{K_i^h}$ and $Z_i^l = \{z_{i,j}^l\}_{j=1}^{K_i^l}$. We leverage the LLM to construct entity-level and slide-level prompts (Section 4.1), and employ a language-guided instance selection strategy to identify tumor-related regions that are most relevant for the discrimination of different cancer subtypes (Section 4.2). Next, the selected features are aggregated to construct entity representations and aligned with subtype-specific attributes to enable fine-grained entity-level classification (Section 4.3). Then, we refine entity representations via the cross-scale graph learning module and subsequently aggregate them to obtain slide-level representations that align with corresponding prompts for slide-level prediction (Section 4.4). Finally, MAPLE jointly optimizes entity-level and slide-level alignment, enabling robust and interpretable few-shot classification (Section 4.5).

### 4.1 LLM-powered Prompt Construction

For the cancer diagnosis from WSIs, pathologists usually combine the observations from key tissue entities (*e.g.,* nuclei, cytoplasm, glands) and the overall context of the entire slide to make decisions [1]. Specifically, for the WSIs observed at high resolution, pathologists analyze cellular components such as nuclear pleomorphism and cytoplasmic features for cancer diagnosis. As to the image with low-resolution, they distinguish different cancer subtypes by examining tissue architecture such as gland formation and tumor-stroma interfaces [20, 16]. Inspired by the aforementioned diagnostic way from the pathologists, we construct multi-scale prompts at both the entity and slide levels to capture discriminative visual attributes associated with specific cancer subtype patterns. Accordingly, the prompt construction process is composed of two parts, *i.e.,* Entity-level Prompt Discovery and Slide-level Prompt Summary, as detailed below.

**Entity-level Prompt Discovery.** Let $\mathcal{C} = \{c_1, c_2, \ldots, c_C\}$ denote the set of cancer subtypes. We construct entities from two scales $\mathcal{E} = \mathcal{E}^h \cup \mathcal{E}^l$, where $\mathcal{E}^h$ and $\mathcal{E}^l$ represent the entities derived from high-resolution and low-resolution images, respectively. For each entity $e \in \mathcal{E}^s$ at scale $s \in \{h, l\}$, we query the LLM to generate two types of textual prompts:
(1) Generic visual description $p_e^{\text{gen},s}$: summarizes the general appearance of entity $e$ at scale $s$;
(2) Subtype-specific attributes $\{p_{e,c}^s\}_{c \in \mathcal{C}}$: describes the attribute of entity $e$ in subtype $c$ at scale $s$.

**Slide-level Prompt Summary.** In prior works [29, 34], slide-level prompts are often defined using class names (*e.g.,* "a WSI of [CLASS]") or with global descriptions directly generated from LLMs. However, such templates fail to reflect the fine-grained entity-level information. To address this, we use the LLM to generate the slide-level prompt $p_c^{\text{slide},s}$ for each cancer subtype $c \in \mathcal{C}$ at scale $s \in \{h, l\}$ by the combination of entity names $\mathcal{E}^s$, and thus can integrate fine-grained entity information into the slide-level representation. More details on prompt construction using LLMs are provided in Appendix A.

### 4.2 Language-guided Instance Selection

Accurate identification of tumor-related regions within gigapixel WSIs is crucial for cancer subtype classification, as these regions contain the most discriminative diagnostic information [7, 27]. Therefore, we propose a language-guided instance selection strategy that can help identify tumor-associated

patches by leveraging the pre-trained vision-language models. Specifically, we first query the LLM with the following instruction: *"What are the visually descriptive characteristics of the tumor-related region in a WSI at high/low resolution?"* Based on this query, the LLM generates region prompts $p_{\text{reg}} = \{p_{\text{reg}}^h, p_{\text{reg}}^l\}$ for high and low resolution, respectively. Then, each prompt is processed by the frozen text encoder $f_t$ to obtain the corresponding text embeddings: $t_{\text{reg}}^h$ and $t_{\text{reg}}^l$. Given the extracted patch features $Z_i^h$ and $Z_i^l$ for slide $X_i$ from different resolutions, we compute the cosine similarity scores between the region text embeddings and patch features from different scales as $S_i^h = \text{sim}(t_{\text{reg}}^h, Z_i^h)$ and $S_i^l = \text{sim}(t_{\text{reg}}^l, Z_i^l)$. Based on these similarity scores, we select the top-$k$ instances with the highest similarity scores at each scale to form the tumor-related instance sets:

$$\tilde{Z}_i^h = \left\{ z_{i,j}^h \mid \text{rank}(S_i^h[j]) < k^h \right\}, \quad \tilde{Z}_i^l = \left\{ z_{i,j}^l \mid \text{rank}(S_i^l[j]) < k^l \right\}, \tag{4}$$

where $k^h$ and $k^l$ denote the number of selected instances at high and low resolutions, respectively.

## 4.3 Entity-guided Attribute-enhanced Classification

To incorporate semantic guidance from entity-level prompts, we encode both the generic descriptions and subtype-specific attributes into learnable embeddings. For each entity $e \in \mathcal{E}^s$ at scale $s \in \{h, l\}$, we prepend a shared set of learnable context vectors $V = \{v_1, \ldots, v_M\}$ to the textual descriptions, forming the learnable prompt tokens $t_e^{\text{gen},s} = \{v_1, \ldots, v_M, p_e^{\text{gen},s}\}$ and $t_{e,c}^s = \{v_1, \ldots, v_M, p_{e,c}^s\}$. These tokens are then encoded by a frozen text encoder $f_t$ to obtain the final prompt embeddings:

$$d_e^{gen,s} = f_t(t_e^{\text{gen},s}), \quad d_{e,c}^s = f_t(t_{e,c}^s). \tag{5}$$

Furthermore, we introduce an Entity-guided Cross-attention module to derive the visual representation of each entity by aggregating instance features. Given the instance set $\tilde{Z}_i^s = \{z_{i,j}^s\}_{j=1}^{K_i^s}$ from slide $X_i$ at scale $s \in \{h, l\}$ and the generic prompt embedding $d_e^{gen,s} \in \mathbb{R}^d$ for entity $e$, the entity-specific feature is computed as:

$$z_{e,i}^s = \text{Norm}\left( \text{softmax}\left( \frac{\mathbf{W}_q d_e^{gen,s}(\mathbf{W}_k \tilde{Z}_i^s)^\top}{\sqrt{d_k}} \right) \mathbf{W}_v \tilde{Z}_i^s \right) + d_e^{gen,s}, \tag{6}$$

where $\mathbf{W}_q, \mathbf{W}_k, \mathbf{W}_v \in \mathbb{R}^{d \times d_k}$ are learnable projection matrices, and $\text{Norm}(\cdot)$ denotes layer normalization. This attention mechanism enables the model to selectively aggregate instances that exhibit strong semantic alignment with entity $e$, yielding a compact and discriminative representation of its visual characteristics within the slide.

For entity-level classification, we compute the cosine similarity between the visual representation $z_{e,i}^s$ of entity $e$ in slide $X_i$ and the corresponding subtype-specific prompt embedding $d_{e,c}^s$ for subtype $c$:

$$\ell_{e,i}^{c,s} = \text{sim}(z_{e,i}^s, d_{e,c}^s). \tag{7}$$

Next, we will integrate $\ell_{e,i}^{c,s}$ with slide-level predictions to derive the final classification results, as detailed in Section 4.5.

## 4.4 Multi-scale Entity Aggregator for Slide-level Classification

In this section, we propose a multi-scale entity aggregator, where we firstly construct a cross-scale entity graph that connects semantically related entities within and across different entity scales to enrich entity representation, and then aggregate the refined entity features to obtain slide-level representations which are aligned with corresponding prompts for slide-level prediction.

**Cross-scale Entity Graph Learning.** Let $\mathcal{Z}_i = \{z_{e,i}^h\}_{e \in \mathcal{E}^h} \cup \{z_{e,i}^l\}_{e \in \mathcal{E}^l}$ denote the set of entity features at different scales for slide $X_i$. We define a graph $\mathcal{G}_i = (\mathcal{V}_i, \mathcal{E}_i)$, where each node $v \in \mathcal{V}_i$ corresponds to an entity feature in $\mathcal{Z}_i$. To capture semantic relationships between entities across different scales, we compute the cosine similarity between features $z_v$ and $z_{v'}$ as $\text{sim}(z_v, z_{v'}) = \frac{z_v^\top z_{v'}}{\|z_v\| \cdot \|z_{v'}\|}$. For each node $v$, its neighborhood $\mathcal{N}(v)$ is defined as the set of top-$k$ most similar nodes:

$$\mathcal{N}(v) = \text{TopK}_{v'}\left(\text{sim}(z_v, z_{v'})\right). \tag{8}$$

Then, the Graph Attention Network (GAT) [35] is applied to propagate information across nodes based on learned attention weights. For each node $v$, the updated entity representation is computed as:

$$\hat{z}_v = \sigma\Big( \sum_{v' \in \mathcal{N}(v)} \alpha_{v,v'} \, \mathbf{W}_g z_{v'} \Big), \tag{9}$$

where $\mathbf{W}_g \in \mathbb{R}^{d' \times d}$ is a learnable weight matrix, $\sigma(\cdot)$ denotes a non-linear activation function (*e.g.*, ReLU), and $\alpha_{v,v'}$ is the attention coefficient:

$$\alpha_{v,v'} = \frac{\exp\big(\mathrm{LeakyReLU}(\mathbf{a}^\top[\mathbf{W}_g z_v \,\|\, \mathbf{W}_g z_{v'}])\big)}{\sum_{u \in \mathcal{N}(v)} \exp\big(\mathrm{LeakyReLU}(\mathbf{a}^\top[\mathbf{W}_g z_v \,\|\, \mathbf{W}_g z_u])\big)}, \tag{10}$$

with $\mathbf{a} \in \mathbb{R}^{2d'}$ being a learnable attention vector, and $\|$ denoting vector concatenation operation.

**Slide-level Representation.** We adopt a gated attention mechanism [23] to derive the slide-level visual representation by aggregating the refined entity features. Let $\hat{\mathbf{H}}_i^s = \{\hat{z}_{e,i}\}_{e \in \mathcal{E}^s} \in \mathbb{R}^{N_s \times d}$ denote the entity features at scale $s$, where $N_s = |\mathcal{E}^s|$. The slide-level feature for scale $s$ can be obtained via the following weighted sum form:

$$z_i^{\mathrm{slide},s} = \boldsymbol{\alpha}^s \hat{\mathbf{H}}_i^s \in \mathbb{R}^{1 \times d}, \tag{11}$$

$$\mathbf{A}^V = \tanh(\hat{\mathbf{H}}_i^s \mathbf{W}_V), \mathbf{A}^U = \sigma(\hat{\mathbf{H}}_i^s \mathbf{W}_U), \boldsymbol{\alpha}^s = \mathrm{softmax}\left((\mathbf{A}^V \odot \mathbf{A}^U)\mathbf{w}\right)^\top, \tag{12}$$

where $\mathbf{W}_V, \mathbf{W}_U \in \mathbb{R}^{d \times d}, \mathbf{w} \in \mathbb{R}^d$ are learnable parameters, $\odot$ denotes element-wise multiplication, and $\boldsymbol{\alpha}^s \in \mathbb{R}^{1 \times N_s}$ represent the normalized attention scores.

**Prompt-based Slide-level Alignment.** For each subtype $c \in \mathcal{C}$ and scale $s \in \{h, l\}$, we construct the slide-level textual prompt by concatenating the learnable context vectors $V$ with the scale-specific slide-level token $p_c^{\mathrm{slide},s}$:

$$t_c^{\mathrm{slide},s} = \{v_1, \ldots, v_M, p_c^{\mathrm{slide},s}\}, \quad d_c^{slide,s} = f_T(t_c^{\mathrm{slide},s}), \tag{13}$$

where $f_T(\cdot)$ denotes the text encoder. The slide-level classification logits for subtype $c$ at scale $s$ can be computed as:

$$\ell_{i,c}^{\mathrm{slide},s} = \mathrm{sim}(z_i^{\mathrm{slide},s}, d_c^{slide,s}). \tag{14}$$

### 4.5 Training Strategy

For each slide $X_i$, we obtain the slide-level logits $\ell_{i,c}^{\mathrm{slide},s}$ (shown in Eq. 14) and entity-level logits $\ell_{e,i}^{c,s}$ (shown in Eq. 7). To integrate both slide-level and fine-grained entity-level logits across different scales, the final classification logits can be computed via weighted combination:

$$\ell_{i,c}^{\mathrm{final}} = \frac{1}{2} \sum_{s \in \{l,h\}} \left[ \lambda \cdot \ell_{i,c}^{\mathrm{slide},s} + (1 - \lambda) \cdot \frac{1}{|\mathcal{E}_i^s|} \sum_{e \in \mathcal{E}_i^s} \ell_{e,i}^{c,s} \right], \tag{15}$$

where $\lambda$ is the hyperparameter that controls the contributions of the slide- and entity-level predictions. Finally, the objective function for our MAPLE is formulated as follows:

$$\mathcal{L} = \mathcal{L}_{\mathrm{CE}}(\ell_{i,c}^{\mathrm{final}}, y_i), \tag{16}$$

where $\mathcal{L}_{CE}$ is the cross-entropy loss defined in Eq. 3 and $y_i \in \mathcal{C}$ is the ground-truth label for slide $X_i$.

## 5 Experiments

**Datasets.** We evaluate MAPLE on three benchmark WSI datasets from The Cancer Genome Atlas (TCGA): TCGA-BRCA, TCGA-RCC, and TCGA-NSCLC. More details for the classification task on each cohort are provided in Appendix B.1. To simulate the few-shot learning scenario in clinical practice, we randomly sample $K$ WSIs per class, where (K = 4, 8, 16 in our implementation).

**Table 1:** Few-shot WSI classification results on TCGA-BRCA, TCGA-RCC, and TCGA-NSCLC datasets under 4-shot, 8-shot, and 16-shot settings. The best results are in **bold**, and the second-best results are underlined.

| Dataset | Methods | TCGA-BRCA | | | TCGA-RCC | | | TCGA-NSCLC | | |
|---|---|---|---|---|---|---|---|---|---|---|
| | | AUC | F1 | ACC | AUC | F1 | ACC | AUC | F1 | ACC |
| 4-shot | ABMIL | 0.665 ± 0.097 | 0.555 ± 0.061 | 0.607 ± 0.068 | 0.876 ± 0.028 | 0.650 ± 0.053 | 0.681 ± 0.053 | 0.626 ± 0.057 | 0.581 ± 0.063 | 0.586 ± 0.061 |
| | TransMIL | 0.646 ± 0.036 | 0.558 ± 0.109 | 0.621 ± 0.139 | 0.881 ± 0.024 | 0.656 ± 0.057 | 0.662 ± 0.063 | 0.629 ± 0.047 | 0.565 ± 0.049 | 0.581 ± 0.041 |
| | GTMIL | 0.679 ± 0.048 | 0.542 ± 0.105 | 0.604 ± 0.136 | 0.883 ± 0.017 | 0.685 ± 0.042 | 0.713 ± 0.048 | 0.663 ± 0.046 | 0.600 ± 0.051 | 0.608 ± 0.040 |
| | WiKG | 0.653 ± 0.029 | 0.536 ± 0.107 | 0.615 ± 0.163 | 0.890 ± 0.030 | 0.658 ± 0.098 | 0.680 ± 0.094 | 0.620 ± 0.050 | 0.563 ± 0.072 | 0.579 ± 0.049 |
| | TOP | 0.652 ± 0.024 | 0.515 ± 0.149 | 0.611 ± 0.186 | 0.854 ± 0.035 | 0.626 ± 0.067 | 0.657 ± 0.069 | 0.624 ± 0.050 | 0.531 ± 0.123 | 0.588 ± 0.061 |
| | ViLa-MIL | 0.663 ± 0.092 | 0.503 ± 0.101 | 0.616 ± 0.159 | 0.878 ± 0.052 | 0.635 ± 0.038 | 0.658 ± 0.037 | 0.629 ± 0.043 | 0.580 ± 0.045 | 0.589 ± 0.037 |
| | MSCPT | 0.678 ± 0.045 | 0.550 ± 0.050 | 0.593 ± 0.077 | 0.872 ± 0.067 | 0.654 ± 0.052 | 0.673 ± 0.062 | 0.626 ± 0.020 | 0.582 ± 0.040 | 0.588 ± 0.038 |
| | FOCUS | 0.703 ± 0.051 | 0.564 ± 0.095 | 0.633 ± 0.148 | 0.880 ± 0.035 | 0.663 ± 0.059 | 0.702 ± 0.054 | 0.713 ± 0.093 | 0.631 ± 0.078 | 0.646 ± 0.067 |
| | **MAPLE** | **0.722 ± 0.063** | **0.594 ± 0.076** | **0.664 ± 0.134** | **0.909 ± 0.020** | **0.705 ± 0.055** | **0.728 ± 0.057** | **0.740 ± 0.056** | **0.663 ± 0.052** | **0.675 ± 0.053** |
| 8-shot | ABMIL | 0.748 ± 0.061 | 0.557 ± 0.098 | 0.593 ± 0.118 | 0.917 ± 0.014 | 0.752 ± 0.028 | 0.768 ± 0.042 | 0.724 ± 0.026 | 0.632 ± 0.031 | 0.634 ± 0.032 |
| | TransMIL | 0.746 ± 0.063 | 0.578 ± 0.035 | 0.630 ± 0.035 | 0.915 ± 0.019 | 0.751 ± 0.040 | 0.765 ± 0.048 | 0.715 ± 0.086 | 0.627 ± 0.127 | 0.631 ± 0.095 |
| | GTMIL | 0.764 ± 0.056 | 0.586 ± 0.092 | 0.633 ± 0.113 | 0.917 ± 0.018 | 0.765 ± 0.036 | 0.781 ± 0.039 | 0.746 ± 0.041 | 0.626 ± 0.072 | 0.644 ± 0.054 |
| | WiKG | 0.709 ± 0.047 | 0.537 ± 0.073 | 0.579 ± 0.100 | 0.909 ± 0.017 | 0.728 ± 0.069 | 0.752 ± 0.065 | 0.740 ± 0.071 | 0.654 ± 0.070 | 0.666 ± 0.066 |
| | TOP | 0.733 ± 0.047 | 0.546 ± 0.059 | 0.580 ± 0.078 | 0.900 ± 0.026 | 0.702 ± 0.067 | 0.736 ± 0.060 | 0.752 ± 0.068 | 0.652 ± 0.038 | 0.663 ± 0.041 |
| | ViLa-MIL | 0.770 ± 0.062 | 0.605 ± 0.065 | 0.653 ± 0.080 | 0.931 ± 0.003 | 0.745 ± 0.032 | 0.777 ± 0.034 | 0.709 ± 0.048 | 0.643 ± 0.042 | 0.649 ± 0.039 |
| | MSCPT | 0.768 ± 0.064 | 0.558 ± 0.067 | 0.596 ± 0.080 | 0.926 ± 0.021 | 0.771 ± 0.038 | 0.792 ± 0.033 | 0.768 ± 0.066 | 0.685 ± 0.072 | 0.692 ± 0.067 |
| | FOCUS | 0.767 ± 0.054 | 0.579 ± 0.100 | 0.616 ± 0.124 | 0.944 ± 0.016 | 0.765 ± 0.043 | 0.783 ± 0.050 | 0.818 ± 0.054 | 0.737 ± 0.066 | 0.739 ± 0.063 |
| | **MAPLE** | **0.786 ± 0.070** | **0.618 ± 0.024** | **0.673 ± 0.018** | **0.957 ± 0.015** | **0.791 ± 0.024** | **0.806 ± 0.024** | **0.855 ± 0.041** | **0.762 ± 0.031** | **0.766 ± 0.030** |
| 16-shot | ABMIL | 0.724 ± 0.048 | 0.587 ± 0.047 | 0.637 ± 0.056 | 0.937 ± 0.012 | 0.757 ± 0.024 | 0.790 ± 0.028 | 0.815 ± 0.041 | 0.747 ± 0.049 | 0.753 ± 0.048 |
| | TransMIL | 0.738 ± 0.092 | 0.597 ± 0.082 | 0.639 ± 0.091 | 0.941 ± 0.029 | 0.812 ± 0.037 | 0.829 ± 0.046 | 0.807 ± 0.057 | 0.740 ± 0.052 | 0.741 ± 0.053 |
| | GTMIL | 0.743 ± 0.051 | 0.619 ± 0.069 | 0.681 ± 0.090 | 0.930 ± 0.009 | 0.803 ± 0.046 | 0.829 ± 0.042 | 0.827 ± 0.040 | 0.750 ± 0.040 | 0.752 ± 0.038 |
| | WiKG | 0.754 ± 0.036 | 0.593 ± 0.054 | 0.637 ± 0.072 | 0.943 ± 0.018 | 0.793 ± 0.037 | 0.816 ± 0.037 | 0.832 ± 0.040 | 0.755 ± 0.039 | 0.756 ± 0.039 |
| | TOP | 0.775 ± 0.025 | 0.636 ± 0.067 | 0.682 ± 0.113 | 0.939 ± 0.015 | 0.769 ± 0.043 | 0.791 ± 0.037 | 0.804 ± 0.066 | 0.738 ± 0.125 | 0.730 ± 0.069 |
| | ViLa-MIL | 0.789 ± 0.026 | 0.651 ± 0.041 | 0.703 ± 0.043 | 0.952 ± 0.007 | 0.797 ± 0.046 | 0.827 ± 0.042 | 0.824 ± 0.055 | 0.757 ± 0.046 | 0.757 ± 0.047 |
| | MSCPT | 0.758 ± 0.043 | 0.642 ± 0.047 | 0.702 ± 0.046 | 0.940 ± 0.013 | 0.813 ± 0.027 | 0.834 ± 0.027 | 0.833 ± 0.027 | 0.765 ± 0.029 | 0.766 ± 0.029 |
| | FOCUS | 0.745 ± 0.052 | 0.633 ± 0.046 | 0.693 ± 0.093 | 0.951 ± 0.008 | 0.826 ± 0.021 | 0.851 ± 0.021 | 0.862 ± 0.056 | 0.781 ± 0.064 | 0.783 ± 0.064 |
| | **MAPLE** | **0.801 ± 0.031** | **0.672 ± 0.076** | **0.735 ± 0.039** | **0.969 ± 0.014** | **0.838 ± 0.034** | **0.867 ± 0.031** | **0.903 ± 0.033** | **0.806 ± 0.060** | **0.810 ± 0.055** |

**Implementation Details.** We utilize CLAM [23] for WSI pre-processing, followed by ViLa-MIL [34] to crop both high-resolution ($10\times$) and low-resolution ($5\times$) WSIs into patches with the size of $256\times256$. We employ PLIP [14] as our vision-language backbone, with a feature dimension of 512 for both visual and textual modalities. GPT-4 [2] is taken as the frozen large language model (LLM). For hyperparameter settings, the the number of entities at each scale $n_k$ is tuned from 4 to 20 with an interval of 4 (Section 4.1), while the number of neighbors for constructing the cross-scale entity graph $n_e = 7$ is tuned from 1 to 13 with interval 2 (Section 4.4). Since WSIs are with different numbers of divided patches, we select the top $\%r$ percentage tumor-related patches for each WSI as the the top-$k$ patches, and we tune $r$ from 0.1 to 1 with interval 0.2. Finally, the weighting parameter $\lambda$ (Section 4.5) for combining entity-level and slide-level predictions is tuned from 0 to 1 with interval 0.1. The model is optimized using AdamW with a learning rate of $1 \times 10^{-4}$ and trained for up to 80 epochs with early stopping based on validation performance. All experiments are conducted using PyTorch 2.0.1 and CUDA 11.7 on Python 3.8 with NVIDIA RTX 3090 GPUs.

**Evaluation Metrics.** The area under the curve (AUC) score, F1 score (F1), and accuracy (ACC) are utilized as the evaluation metrics in our experiment. We conduct five-fold cross-validation and the mean and standard deviation are calculated according to the results of all folds.

## 5.1 Main Results

We compare MAPLE with SOTA MIL based methods including ABMIL [15], TransMIL [32], GTMIL [42], and WiKG-MIL [19], as well as SOTA prompt based methods in few-shot WSI classification methods like TOP [29], ViLa-MIL [34], MSCPT [11] and FOCUS [10]. PLIP [14] is applied to extract both visual and textual features for all methods. We compare MAPLE with these methods across three datasets under three few-shot settings (4-shot, 8-shot, and 16-shot), as shown in Tab. 1. We provide additional comparison results using different vision-language models (*i.e.,* CLIP [30] and CONCH [24]) in Appendix C.1 and C.3 and further comparisons of model complexity and efficiency in Appendix B.3.

**TCGA-BRCA (2 classes).** On the TCGA-BRCA dataset, our proposed MAPLE consistently outperforms all baseline methods across all few-shot settings. In the 16-shot scenario, MAPLE achieves an AUC of 80.1%, F1 of 67.2% and ACC of 73.5%, surpassing the second-best performing ViLa-MIL with an improvement of 1.2% in AUC, 2.1% in F1 and 3.2% in ACC. This advantage is

**Table 2:** Effects of different levels and entity scales under the 16-shot setting.

| Methods | TCGA-BRCA | | | TCGA-RCC | | | TCGA-NSCLC | | |
|---|---|---|---|---|---|---|---|---|---|
| | AUC | F1 | ACC | AUC | F1 | ACC | AUC | F1 | ACC |
| MAPLE-Low | 0.790 ± 0.052 | 0.640 ± 0.079 | 0.704 ± 0.101 | 0.949 ± 0.009 | 0.796 ± 0.037 | 0.832 ± 0.030 | 0.866 ± 0.045 | 0.780 ± 0.046 | 0.782 ± 0.045 |
| MAPLE-High | 0.779 ± 0.046 | 0.651 ± 0.057 | 0.729 ± 0.080 | 0.956 ± 0.016 | 0.817 ± 0.044 | 0.842 ± 0.041 | 0.875 ± 0.050 | 0.789 ± 0.049 | 0.791 ± 0.048 |
| MAPLE-Entity | 0.792 ± 0.052 | 0.653 ± 0.077 | 0.714 ± 0.104 | 0.960 ± 0.014 | 0.815 ± 0.032 | 0.846 ± 0.031 | 0.879 ± 0.060 | 0.787 ± 0.045 | 0.796 ± 0.044 |
| MAPLE-Slide | 0.789 ± 0.036 | 0.644 ± 0.050 | 0.702 ± 0.076 | 0.951 ± 0.011 | 0.819 ± 0.041 | 0.850 ± 0.037 | 0.891 ± 0.033 | 0.777 ± 0.046 | 0.785 ± 0.043 |
| **MAPLE** | **0.801 ± 0.031** | **0.672 ± 0.076** | **0.735 ± 0.039** | **0.969 ± 0.014** | **0.838 ± 0.034** | **0.867 ± 0.031** | **0.903 ± 0.033** | **0.806 ± 0.060** | **0.810 ± 0.055** |

**Table 3:** Ablation study of each component under the 16-shot setting. "w/o Selection" refers to MAPLE without language-guided instance selection step, "w/o EGCA" refers to MAPLE without entity-guided cross-attention module, and "w/o Graph" refers to MAPLE without the module of cross-scale entity graph learning.

| Methods | TCGA-BRCA | | | TCGA-RCC | | | TCGA-NSCLC | | |
|---|---|---|---|---|---|---|---|---|---|
| | AUC | F1 | ACC | AUC | F1 | ACC | AUC | F1 | ACC |
| MAPLE (w/o Selection) | 0.792 ± 0.059 | 0.655 ± 0.040 | 0.710 ± 0.045 | 0.962 ± 0.015 | 0.828 ± 0.056 | 0.859 ± 0.053 | 0.885 ± 0.064 | 0.786 ± 0.058 | 0.791 ± 0.057 |
| MAPLE (w/o EGCA) | 0.784 ± 0.019 | 0.651 ± 0.043 | 0.710 ± 0.061 | 0.959 ± 0.015 | 0.820 ± 0.031 | 0.852 ± 0.032 | 0.879 ± 0.042 | 0.782 ± 0.048 | 0.790 ± 0.047 |
| MAPLE (w/o Graph) | 0.793 ± 0.038 | 0.658 ± 0.073 | 0.716 ± 0.101 | 0.957 ± 0.011 | 0.823 ± 0.036 | 0.855 ± 0.031 | 0.887 ± 0.039 | 0.791 ± 0.047 | 0.792 ± 0.046 |
| **MAPLE** | **0.801 ± 0.031** | **0.672 ± 0.076** | **0.735 ± 0.039** | **0.969 ± 0.014** | **0.838 ± 0.034** | **0.867 ± 0.031** | **0.903 ± 0.033** | **0.806 ± 0.060** | **0.810 ± 0.055** |

maintained with extremely limited training data. In the challenging 4-shot setting, MAPLE achieves an AUC of 72.2%, F1 of 59.4%, and ACC of 66.4%, demonstrating significant improvements over the second-best method FOCUS (AUC: 70.3%, F1: 56.4%, ACC: 63.3%).

**TCGA-RCC (3 classes).**  For the multi-class TCGA-RCC dataset, MAPLE delivers exceptional performance across all metrics and settings. In the 16-shot scenario, MAPLE achieves the highest AUC (96.9%), F1 (83.8%), and ACC (86.7%), showing significant improvements over the second-best method FOCUS. As the number of shots decreases to 8 and 4, MAPLE maintains its superior performance with AUC scores of 95.7% and 90.9%, respectively.

**TCGA-NSCLC (2 classes).**  For the TCGA-NSCLC dataset, MAPLE again achieves superior performance across all metrics, with AUCs of 74.0%, 85.5%, and 90.3% in the 4, 8, and 16-shot settings respectively, consistently outperforming FOCUS (AUC: 71.3%, 81.8%, and 86.2%).

Overall, MAPLE consistently outperforms all baselines across three cancer datasets under different few-shot settings, validating the effectiveness of MAPLE in few-shot WSI classification.

## 5.2  Ablation Study

We conduct comprehensive ablation studies to investigate the effectiveness of each component in MAPLE. The more detailed analysis of the hyperparameters (*i.e., $n_e$, $n_k$, $r$, and $\lambda$* mentioned in *implementation details*) and the impact of different large language models is provided in Appendix D.

**Impact of Combining Slide-level and Entity-level Logits.**  To evaluate the effectiveness of combining both slide-level and entity-level logits for cancer diagnosis, we compare MAPLE with its competitors that only using entity-level (MAPLE-Entity) and slide-level (MAPLE-Slide) logits under the 16-shot setting in Tab. 2. Notably, the entity-level information can achieve comparable or even slightly superior performance to the results using slide-level logits, demonstrating the effectiveness of modeling fine-grained histological entities and their subtype-specific attributes for cancer diagnosis. In addition, it is obviously that our MAPLE performs better than MAPLE-Entity and MAPLE-Slide, which confirms the complementary nature of entity-level and slide-level information for cancer subtype classification. We also discuss the effects of parameter $\lambda$ that is applied to balance the contribution of slide-level and entity-level logits in Appendix D.4.

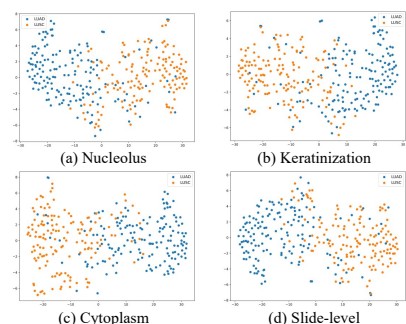

**Figure 3:** t-SNE results of entity-level (a–c) and slide-level (d) embeddings on the TCGA-NSCLC dataset.

To further demonstrate the discriminative power of entity-level and slide-level representations, we visualize the learned representations from the entities of Nucleolus (Fig. 3 (a)), Keratinization (Fig. 3 (b)) and Cytoplasm (Fig. 3 (c)), and slide-level embeddings ((Fig. 3 (d))) on the TCGA-NSCLC dataset using t-SNE. As shown in Fig. 3, all representations exhibit clear separation among different cancer subtypes, confirming their ability to capture subtype-specific patterns.

**Impact of Multi-scale Entities.**  To evaluate the effectiveness of integrating multi-scale entities for few-shot WSI classification, we compare MAPLE with its two variants that rely solely on entities

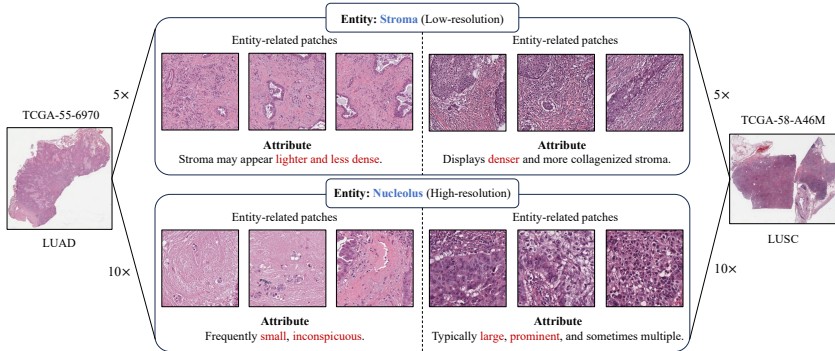

**Figure 4:** Visualization of entity-relevant patches selected by the entity-guided cross-attention module for lung adenocarcinoma (LUAD) and lung squamous cell carcinoma (LUSC) on the TCGA-NSCLC dataset. Top rows show patches and their corresponding entity attributes (*e.g.,* stroma) at low resolution, while bottom rows show patches and their corresponding entity attributes (*e.g.,* nucleoli) at high resolution.

extracted from high-resolution (MAPLE-High) or low-resolution (MAPLE-Low) images. As shown in Tab. 2, MAPLE yields higher classification results to MAPLE-High and MAPLE-Low, which validates the superiority of integrating multi-scale entities for cancer subtype classification.

**Impact of Each Component.** We further ablate three key components of MAPLE, *i.e.,* language-guided instance selection module, entity-guided cross-attention module and cross-scale entity graph learning module, to assess their contributions. As shown in Tab. 3, each ablation study achieves inferior performance to MAPLE, indicating that each component contributes positively to the results.

## 5.3 Visualization Results

To validate entity-level interpretability of MAPLE, we visualize instances identified by our entity-guided cross-attention module on the TCGA-NSCLC dataset. Fig. 4 presents patches selected using generic entity descriptions, which exhibit subtype-specific characteristics matching their corresponding attribute prompts. For example, for the image patch at low resolution, stroma-related patches from lung adenocarcinoma (LUAD) appear lighter and less dense, while those from lung squamous cell carcinoma (LUSC) display denser and more collagenized patterns. Similarly, for the image patch at high resolution, nucleolus-focused patches from LUAD contain small, inconspicuous nucleoli, whereas those from LUSC exhibit large, prominent nucleoli. These visualizations confirm that MAPLE effectively captures the relationships between entity-related patches and their subtype-specific phenotypic attributes described in our LLM-generated prompts, providing visual evidence for our classification decisions. More visualization results on the TCGA-BRCA and TCGA-RCC datasets are provided in Appendix E.

## 6 Conclusion

In this paper, we introduce MAPLE, a hierarchical prompt learning framework for few-shot WSI classification that explicitly models multi-scale histological entities and their phenotypic attributes. By bridging the semantic gap between fine-grained visual details and textual descriptions through entity-level and slide-level alignment, MAPLE provides both enhanced classification accuracy and improved interpretability for cancer diagnosis. Extensive experiments across three cancer datasets demonstrate that our approach consistently outperforms state-of-the-art methods. The hierarchical nature of MAPLE aligns well with pathologists' diagnostic workflow, offering a promising direction for computer-aided diagnosis in computational pathology.

## Acknowledgment

This work is supported by the National Natural Science Foundation of China (Nos. 62136004, 62572239, 62272226, 62571019, 62176013, 62402219), the National Key R&D Program of China (No. 2023YFF1204803), the Key Research and Development Plan of Jiangsu Province (No. BE2022842), Natural Science Foundation of Jiangsu Province(No. BC20241382) and Jiangsu Funding Program for Excellent Postdoctoral Talent (No. 2024ZB020).

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

# NeurIPS Paper Checklist

1. **Claims**

   Question: Do the main claims made in the abstract and introduction accurately reflect the paper's contributions and scope?

   Answer: [Yes]

   Justification: The contributions and scope of this paper are claimed in the abstract and introduction.

   Guidelines:

   - The answer NA means that the abstract and introduction do not include the claims made in the paper.
   - The abstract and/or introduction should clearly state the claims made, including the contributions made in the paper and important assumptions and limitations. A No or NA answer to this question will not be perceived well by the reviewers.
   - The claims made should match theoretical and experimental results, and reflect how much the results can be expected to generalize to other settings.
   - It is fine to include aspirational goals as motivation as long as it is clear that these goals are not attained by the paper.

2. **Limitations**

   Question: Does the paper discuss the limitations of the work performed by the authors?

   Answer: [Yes]

   Justification: We discuss the limitations of our paper in Appendix F.1.

   Guidelines:

   - The answer NA means that the paper has no limitation while the answer No means that the paper has limitations, but those are not discussed in the paper.
   - The authors are encouraged to create a separate "Limitations" section in their paper.
   - The paper should point out any strong assumptions and how robust the results are to violations of these assumptions (e.g., independence assumptions, noiseless settings, model well-specification, asymptotic approximations only holding locally). The authors should reflect on how these assumptions might be violated in practice and what the implications would be.
   - The authors should reflect on the scope of the claims made, e.g., if the approach was only tested on a few datasets or with a few runs. In general, empirical results often depend on implicit assumptions, which should be articulated.
   - The authors should reflect on the factors that influence the performance of the approach. For example, a facial recognition algorithm may perform poorly when image resolution is low or images are taken in low lighting. Or a speech-to-text system might not be used reliably to provide closed captions for online lectures because it fails to handle technical jargon.
   - The authors should discuss the computational efficiency of the proposed algorithms and how they scale with dataset size.
   - If applicable, the authors should discuss possible limitations of their approach to address problems of privacy and fairness.
   - While the authors might fear that complete honesty about limitations might be used by reviewers as grounds for rejection, a worse outcome might be that reviewers discover limitations that aren't acknowledged in the paper. The authors should use their best judgment and recognize that individual actions in favor of transparency play an important role in developing norms that preserve the integrity of the community. Reviewers will be specifically instructed to not penalize honesty concerning limitations.

3. **Theory assumptions and proofs**

   Question: For each theoretical result, does the paper provide the full set of assumptions and a complete (and correct) proof?

   Answer: [NA]

Justification: This paper does not include theoretical results.

Guidelines:

- The answer NA means that the paper does not include theoretical results.
- All the theorems, formulas, and proofs in the paper should be numbered and cross-referenced.
- All assumptions should be clearly stated or referenced in the statement of any theorems.
- The proofs can either appear in the main paper or the supplemental material, but if they appear in the supplemental material, the authors are encouraged to provide a short proof sketch to provide intuition.
- Inversely, any informal proof provided in the core of the paper should be complemented by formal proofs provided in appendix or supplemental material.
- Theorems and Lemmas that the proof relies upon should be properly referenced.

4. **Experimental result reproducibility**

Question: Does the paper fully disclose all the information needed to reproduce the main experimental results of the paper to the extent that it affects the main claims and/or conclusions of the paper (regardless of whether the code and data are provided or not)?

Answer: [Yes]

Justification: The experimental settings and details are provided in the experiment section.

Guidelines:

- The answer NA means that the paper does not include experiments.
- If the paper includes experiments, a No answer to this question will not be perceived well by the reviewers: Making the paper reproducible is important, regardless of whether the code and data are provided or not.
- If the contribution is a dataset and/or model, the authors should describe the steps taken to make their results reproducible or verifiable.
- Depending on the contribution, reproducibility can be accomplished in various ways. For example, if the contribution is a novel architecture, describing the architecture fully might suffice, or if the contribution is a specific model and empirical evaluation, it may be necessary to either make it possible for others to replicate the model with the same dataset, or provide access to the model. In general. releasing code and data is often one good way to accomplish this, but reproducibility can also be provided via detailed instructions for how to replicate the results, access to a hosted model (e.g., in the case of a large language model), releasing of a model checkpoint, or other means that are appropriate to the research performed.
- While NeurIPS does not require releasing code, the conference does require all submissions to provide some reasonable avenue for reproducibility, which may depend on the nature of the contribution. For example
  (a) If the contribution is primarily a new algorithm, the paper should make it clear how to reproduce that algorithm.
  (b) If the contribution is primarily a new model architecture, the paper should describe the architecture clearly and fully.
  (c) If the contribution is a new model (e.g., a large language model), then there should either be a way to access this model for reproducing the results or a way to reproduce the model (e.g., with an open-source dataset or instructions for how to construct the dataset).
  (d) We recognize that reproducibility may be tricky in some cases, in which case authors are welcome to describe the particular way they provide for reproducibility. In the case of closed-source models, it may be that access to the model is limited in some way (e.g., to registered users), but it should be possible for other researchers to have some path to reproducing or verifying the results.

5. **Open access to data and code**

Question: Does the paper provide open access to the data and code, with sufficient instructions to faithfully reproduce the main experimental results, as described in supplemental material?

Answer: [Yes]

Justification: The datasets are public. The source code will be publicly available upon acceptance of the paper.

Guidelines:

- The answer NA means that paper does not include experiments requiring code.
- Please see the NeurIPS code and data submission guidelines (`https://nips.cc/public/guides/CodeSubmissionPolicy`) for more details.
- While we encourage the release of code and data, we understand that this might not be possible, so "No" is an acceptable answer. Papers cannot be rejected simply for not including code, unless this is central to the contribution (e.g., for a new open-source benchmark).
- The instructions should contain the exact command and environment needed to run to reproduce the results. See the NeurIPS code and data submission guidelines (`https://nips.cc/public/guides/CodeSubmissionPolicy`) for more details.
- The authors should provide instructions on data access and preparation, including how to access the raw data, preprocessed data, intermediate data, and generated data, etc.
- The authors should provide scripts to reproduce all experimental results for the new proposed method and baselines. If only a subset of experiments are reproducible, they should state which ones are omitted from the script and why.
- At submission time, to preserve anonymity, the authors should release anonymized versions (if applicable).
- Providing as much information as possible in supplemental material (appended to the paper) is recommended, but including URLs to data and code is permitted.

6. **Experimental setting/details**

Question: Does the paper specify all the training and test details (e.g., data splits, hyper-parameters, how they were chosen, type of optimizer, etc.) necessary to understand the results?

Answer: [Yes]

Justification: The experimental settings and details are provided in the experiment section.

Guidelines:

- The answer NA means that the paper does not include experiments.
- The experimental setting should be presented in the core of the paper to a level of detail that is necessary to appreciate the results and make sense of them.
- The full details can be provided either with the code, in appendix, or as supplemental material.

7. **Experiment statistical significance**

Question: Does the paper report error bars suitably and correctly defined or other appropriate information about the statistical significance of the experiments?

Answer: [Yes]

Justification: We report the mean and standard deviation of the results based on five-fold cross-validation across all major experiments, ensuring statistical robustness.

Guidelines:

- The answer NA means that the paper does not include experiments.
- The authors should answer "Yes" if the results are accompanied by error bars, confidence intervals, or statistical significance tests, at least for the experiments that support the main claims of the paper.
- The factors of variability that the error bars are capturing should be clearly stated (for example, train/test split, initialization, random drawing of some parameter, or overall run with given experimental conditions).
- The method for calculating the error bars should be explained (closed form formula, call to a library function, bootstrap, etc.)
- The assumptions made should be given (e.g., Normally distributed errors).

- It should be clear whether the error bar is the standard deviation or the standard error of the mean.
- It is OK to report 1-sigma error bars, but one should state it. The authors should preferably report a 2-sigma error bar than state that they have a 96% CI, if the hypothesis of Normality of errors is not verified.
- For asymmetric distributions, the authors should be careful not to show in tables or figures symmetric error bars that would yield results that are out of range (e.g. negative error rates).
- If error bars are reported in tables or plots, The authors should explain in the text how they were calculated and reference the corresponding figures or tables in the text.

8. **Experiments compute resources**

Question: For each experiment, does the paper provide sufficient information on the computer resources (type of compute workers, memory, time of execution) needed to reproduce the experiments?

Answer: [Yes]

Justification: They are included in Section 5 and Appendix B.3.

Guidelines:

- The answer NA means that the paper does not include experiments.
- The paper should indicate the type of compute workers CPU or GPU, internal cluster, or cloud provider, including relevant memory and storage.
- The paper should provide the amount of compute required for each of the individual experimental runs as well as estimate the total compute.
- The paper should disclose whether the full research project required more compute than the experiments reported in the paper (e.g., preliminary or failed experiments that didn't make it into the paper).

9. **Code of ethics**

Question: Does the research conducted in the paper conform, in every respect, with the NeurIPS Code of Ethics https://neurips.cc/public/EthicsGuidelines?

Answer: [Yes]

Justification: The research conducted in the paper conform with the NeurIPS Code of Ethics.

Guidelines:

- The answer NA means that the authors have not reviewed the NeurIPS Code of Ethics.
- If the authors answer No, they should explain the special circumstances that require a deviation from the Code of Ethics.
- The authors should make sure to preserve anonymity (e.g., if there is a special consideration due to laws or regulations in their jurisdiction).

10. **Broader impacts**

Question: Does the paper discuss both potential positive societal impacts and negative societal impacts of the work performed?

Answer: [Yes]

Justification: We discuss the boarder impact of our paper in Appendix F.2.

Guidelines:

- The answer NA means that there is no societal impact of the work performed.
- If the authors answer NA or No, they should explain why their work has no societal impact or why the paper does not address societal impact.
- Examples of negative societal impacts include potential malicious or unintended uses (e.g., disinformation, generating fake profiles, surveillance), fairness considerations (e.g., deployment of technologies that could make decisions that unfairly impact specific groups), privacy considerations, and security considerations.

- The conference expects that many papers will be foundational research and not tied to particular applications, let alone deployments. However, if there is a direct path to any negative applications, the authors should point it out. For example, it is legitimate to point out that an improvement in the quality of generative models could be used to generate deepfakes for disinformation. On the other hand, it is not needed to point out that a generic algorithm for optimizing neural networks could enable people to train models that generate Deepfakes faster.
- The authors should consider possible harms that could arise when the technology is being used as intended and functioning correctly, harms that could arise when the technology is being used as intended but gives incorrect results, and harms following from (intentional or unintentional) misuse of the technology.
- If there are negative societal impacts, the authors could also discuss possible mitigation strategies (e.g., gated release of models, providing defenses in addition to attacks, mechanisms for monitoring misuse, mechanisms to monitor how a system learns from feedback over time, improving the efficiency and accessibility of ML).

11. **Safeguards**

Question: Does the paper describe safeguards that have been put in place for responsible release of data or models that have a high risk for misuse (e.g., pretrained language models, image generators, or scraped datasets)?

Answer: [NA]

Justification: Our paper does not include generative models and typically uses open-source datasets for training and evaluation.

Guidelines:

- The answer NA means that the paper poses no such risks.
- Released models that have a high risk for misuse or dual-use should be released with necessary safeguards to allow for controlled use of the model, for example by requiring that users adhere to usage guidelines or restrictions to access the model or implementing safety filters.
- Datasets that have been scraped from the Internet could pose safety risks. The authors should describe how they avoided releasing unsafe images.
- We recognize that providing effective safeguards is challenging, and many papers do not require this, but we encourage authors to take this into account and make a best faith effort.

12. **Licenses for existing assets**

Question: Are the creators or original owners of assets (e.g., code, data, models), used in the paper, properly credited and are the license and terms of use explicitly mentioned and properly respected?

Answer: [Yes]

Justification: We cite the original papers that produced the code package and datasets.

Guidelines:

- The answer NA means that the paper does not use existing assets.
- The authors should cite the original paper that produced the code package or dataset.
- The authors should state which version of the asset is used and, if possible, include a URL.
- The name of the license (e.g., CC-BY 4.0) should be included for each asset.
- For scraped data from a particular source (e.g., website), the copyright and terms of service of that source should be provided.
- If assets are released, the license, copyright information, and terms of use in the package should be provided. For popular datasets, `paperswithcode.com/datasets` has curated licenses for some datasets. Their licensing guide can help determine the license of a dataset.
- For existing datasets that are re-packaged, both the original license and the license of the derived asset (if it has changed) should be provided.

- If this information is not available online, the authors are encouraged to reach out to the asset's creators.

13. **New assets**

Question: Are new assets introduced in the paper well documented and is the documentation provided alongside the assets?

Answer: [NA]

Justification: Although we will submit the code in the supplementary materials, we will continue to improve the codebase and make it publicly available after the paper is officially accepted. Currently, we have not released any new assets.

Guidelines:

- The answer NA means that the paper does not release new assets.
- Researchers should communicate the details of the dataset/code/model as part of their submissions via structured templates. This includes details about training, license, limitations, etc.
- The paper should discuss whether and how consent was obtained from people whose asset is used.
- At submission time, remember to anonymize your assets (if applicable). You can either create an anonymized URL or include an anonymized zip file.

14. **Crowdsourcing and research with human subjects**

Question: For crowdsourcing experiments and research with human subjects, does the paper include the full text of instructions given to participants and screenshots, if applicable, as well as details about compensation (if any)?

Answer: [NA]

Justification: This paper does not involve crowdsourcing nor research with human subjects.

Guidelines:

- The answer NA means that the paper does not involve crowdsourcing nor research with human subjects.
- Including this information in the supplemental material is fine, but if the main contribution of the paper involves human subjects, then as much detail as possible should be included in the main paper.
- According to the NeurIPS Code of Ethics, workers involved in data collection, curation, or other labor should be paid at least the minimum wage in the country of the data collector.

15. **Institutional review board (IRB) approvals or equivalent for research with human subjects**

Question: Does the paper describe potential risks incurred by study participants, whether such risks were disclosed to the subjects, and whether Institutional Review Board (IRB) approvals (or an equivalent approval/review based on the requirements of your country or institution) were obtained?

Answer: [NA]

Justification: This paper does not involve crowdsourcing nor research with human subjects.

Guidelines:

- The answer NA means that the paper does not involve crowdsourcing nor research with human subjects.
- Depending on the country in which research is conducted, IRB approval (or equivalent) may be required for any human subjects research. If you obtained IRB approval, you should clearly state this in the paper.
- We recognize that the procedures for this may vary significantly between institutions and locations, and we expect authors to adhere to the NeurIPS Code of Ethics and the guidelines for their institution.
- For initial submissions, do not include any information that would break anonymity (if applicable), such as the institution conducting the review.

16. **Declaration of LLM usage**

    Question: Does the paper describe the usage of LLMs if it is an important, original, or non-standard component of the core methods in this research? Note that if the LLM is used only for writing, editing, or formatting purposes and does not impact the core methodology, scientific rigorousness, or originality of the research, declaration is not required.

    Answer: [NA]

    Justification: The core method development in this research does not involve LLMs as any important, original, or non-standard components.

    Guidelines:

    - The answer NA means that the core method development in this research does not involve LLMs as any important, original, or non-standard components.
    - Please refer to our LLM policy (`https://neurips.cc/Conferences/2025/LLM`) for what should or should not be described.

*Appendix for*

## "MAPLE: Multi-scale Attribute-enhanced Prompt Learning for Few-shot Whole Slide Image Classification"

Appendix organization:

---

---

## A Prompt Construction

### A.1 Algorithms

The detailed prompt construction process described in Section 4.1 can be conducted by using an LLM (*e.g.,* GPT-4) as outlined in Algorithm 1. This includes iterative entity-level prompt discovery, and slide-level prompt synthesis. Specifically, we begin by iteratively expanding a pool of histologically meaningful entities, where each selected entity highlights distinct structures relevant to subtype classification. For each entity, the LLM is queried to generate both a general visual description and subtype-specific morphological characteristics. These outputs are formatted into structured

---

**Algorithm 1** LLM-powered Prompt Construction

---

**Require:** Subtype list $\mathcal{C} = \{c_1, c_2, \ldots, c_C\}$; maximum number of entities $N_e$
**Ensure:** Entity-level prompts $\mathcal{P}_{\text{entity}}^s$, slide-level prompts $\mathcal{P}_{\text{slide}}^s$
 1: Initialize entity pool $\mathcal{E}^s \leftarrow \emptyset$, prompts $\mathcal{P}_{\text{entity}}^s \leftarrow \emptyset$
 2: **while** $|\mathcal{E}^s| < N_e$ **do**
 3:     $e \leftarrow$ QUERYLLM(''Suggest a discriminative histological entity not in $\mathcal{E}$ that helps distinguish subtypes in $\mathcal{C}$ at scale $s$'')
 4:     Add $e$ to $\mathcal{E}^s$
 5:     $p_e^{\text{gen, s}} \leftarrow$ QUERYLLM(''Describe generic visual characteristics of $e$ at scale $s$'')
 6:     **for** each subtype $c \in \mathcal{C}$ **do**
 7:         $p_{e,c}^s \leftarrow$ QUERYLLM(''Describe how $e$ appears in subtype $c$ at scale $s$'')
 8:     **end for**
 9:     $p_e^s \leftarrow$ FORMATPROMPT($e, p_e^{\text{gen, s}}, \{p_{e,c}^s\}_{c \in \mathcal{C}}$)
10:     Append $p_e^s$ to $\mathcal{P}_{\text{entity}}^s$
11: **end while**
12: Initialize $\mathcal{P}_{\text{slide}}^s \leftarrow \emptyset$
13: **for** each subtype $c \in \mathcal{C}$ **do**
14:     $context_c \leftarrow$ COLLECTENTITIES($c, \mathcal{P}_{\text{entity}}^s$)
15:     $p_c^{\text{slide, s}} \leftarrow$ QUERYLLM("Describe a WSI of $c$ at scale $s$ based on: $context_c$)
16:     Append $p_c^{\text{slide, s}}$ to $\mathcal{P}_{\text{slide}}^s$
17: **end for**
18: **return** $\mathcal{P}_{\text{entity}}^s, \mathcal{P}_{\text{slide}}^s$

---

prompts used for fine-grained entity-level alignment. After constructing the entity-level prompts, we generate slide-level prompts by prompting the LLM to compose holistic WSI descriptions. These are conditioned on the entity attributes associated with each subtype, thereby enriching slide-level prompts with fine-grained context. This approach enables our model to jointly capture both entity-level and slide-level semantic cues essential for few-shot classification.

### A.2 Examples

We provide examples of the constructed prompts at low and high resolutions on the TCGA-NSCLC in Fig. 5 and 6, respectively.

## B Experimental Details

### B.1 Dataset Details

**TCGA-BRCA.** This dataset contains 1,054 whole-slide images (WSIs) of breast invasive carcinoma (BRCA) collected from TCGA[2]. It comprises 843 slides of invasive ductal carcinoma (IDC) and 211 slides of invasive lobular carcinoma (ILC).

**TCGA-RCC.** This dataset includes 873 WSIs of renal cell carcinoma (RCC) obtained from TCGA. It consists of 455 chromophobe RCC (CHRCC), 121 papillary RCC (PRCC), and 297 clear cell RCC (CCRCC) slides.

**TCGA-NSCLC.** This dataset comprises 1,039 WSIs of non-small cell lung cancer (NSCLC) from TCGA, including 530 slides of lung adenocarcinoma (LUAD) and 509 slides of lung squamous cell carcinoma (LUSC).

---

[2]https://portal.gdc.cancer.gov

Examples

```
"low": {
  "entities": [
    {
      "name": "Architecture",
      "general_feature": "Overall pattern or arrangement of tumor cells and structures within the tumor region visible at low magnification.",
      "attributes": {
        "lung adenocarcinoma": "Shows glandular, acinar, or papillary arrangements with well-formed gland-like spaces and frequent areas of lepidic (growth along alveolar spaces) pattern.",
        "lung squamous cell carcinoma": "Displays solid nests, sheets, or islands of tumor cells with evidence of keratinization or central necrosis (keratin pearls), lacking glandular or papillary structures."
      }
    },
    {
      "name": "Stroma",
      "general_feature": "The connective tissue and extracellular matrix surrounding tumor cell nests; appears as fibrous or desmoplastic regions.",
      "attributes": {
        "lung adenocarcinoma": "Typically features prominent desmoplastic reaction with loose, myxoid, or fibrous stroma separating irregular glandular structures; stroma may appear lighter and less dense at low magnification.",
        "lung squamous cell carcinoma": "Displays denser and more collagenized stroma, often with fewer desmoplastic changes and a tendency for stromal tissue to tightly encase solid sheets or nests of tumor cells."
      }
    },
    {
      "name": "Lumina",
      "general_feature": "Round to oval empty spaces or gland-like areas within tumor cell nests seen under low magnification.",
      "attributes": {
        "lung adenocarcinoma": "Frequent presence of well-formed glandular lumina, sometimes with mucin, demonstrating distinct glandular differentiation in the tumor nests.",
        "lung squamous cell carcinoma": "Rare to absent true lumina; if present, are irregular, slit-like, or represent necrotic debris rather than glandular differentiation, with tumor nests generally appearing solid."
      }
    },
......
```

**Figure 5:** Example of the constructed prompts at low resolution on the TCGA-NSCLC dataset.

```
"high": {
  "entities": [
    {
      "name": "Cytoplasmic Border",
      "general_feature": "The outline or edge of the cytoplasm surrounding the nucleus,
often seen as the boundary between adjacent tumor cells.",
      "attributes": {
        "lung adenocarcinoma": "Cytoplasmic borders are generally indistinct and poorly
defined, with cells often demonstrating overlapping or poorly separated cytoplasm. Cell
cohesion is low.",
        "lung squamous cell carcinoma": "Cytoplasmic borders are well-delineated and
sharp, with clear cell-to-cell boundaries. Tumor cells exhibit strong cohesion and
polygonal shapes."
      }
    },
    {
      "name": "Nucleolus",
      "general_feature": "Prominent, round to oval intranuclear structure, appears as a
distinct spot within the nucleus under high magnification.",
      "attributes": {
        "lung adenocarcinoma": "Frequently small, inconspicuous or variably prominent;
may be solitary or few per nucleus with generally fine chromatin background.",
        "lung squamous cell carcinoma": "Typically large, prominent, and sometimes
multiple; often eosinophilic with hyperchromatic surrounding chromatin and increased
irregularity."
      }
    },
    {
      "name": "Keratinization",
      "general_feature": "Presence and appearance of eosinophilic, dense, concentric
cytoplasmic material or whorls within tumor cells.",
      "attributes": {
        "lung adenocarcinoma": "Typically absent; tumor cells rarely show keratin
production or keratin pearls.",
        "lung squamous cell carcinoma": "Frequently present; keratin pearls and dense
eosinophilic cytoplasmic material are prominent, often forming concentric whorled
structures."
      }
    },
......
```

**Figure 6:** Example of the constructed prompts at high resolution on the TCGA-NSCLC dataset.

## B.2 Descriptions of Compared Methods

**ABMIL [15].** ABMIL introduces an attention-based multiple instance learning framework that assigns importance weights to individual instances, enabling the model to aggregate instance features into a slide-level representation with improved interpretability.

**TransMIL [32].** TransMIL employs a transformer architecture to capture both morphological features and spatial relationships among instances, enabling effective performance across binary and multi-class classification tasks.

**GTMIL [42].** GTMIL proposes a graph-based vision transformer that integrates structural WSI representations with transformer-based feature modeling for whole slide image classification.

**WiKG [19].** WiKG conceptualizes a WSI as a dynamic knowledge graph, where neighbor instances and edge embeddings are dynamically constructed based on semantic relationships, enabling context-aware graph reasoning.

**TOP [29].** TOP introduces a two-level prompt learning strategy that incorporates linguistic prior knowledge to guide both instance- and slide-level feature aggregation within a vision-language modeling framework.

**ViLa-MIL [34].** ViLa-MIL designs dual-scale visual prompts using a frozen LLM to enhance vision-language alignment, effectively boosting few-shot performance in pathology tasks.

**MSCPT [11].** MSCPT presents a graph-based prompt tuning module to encode contextual dependencies across WSI patches, followed by a cross-guided non-parametric aggregation scheme for WSI-level representation learning.

**FOCUS [10].** FOCUS integrates foundation models and language-guided patch selection to prioritize diagnostically relevant regions, enabling focused and efficient analysis in weakly supervised settings.

Table 4: Comparisons of model complexity and efficiency. We report the number of trainable parameters (MB), inference time per slide (ms), and training time per epoch (s) on the TCGA-NSCLC dataset under the 16-shot setting.

| Methods | Trainable Params | Inference Time (ms / slide) | Training Time (s / epoch) |
|---|---|---|---|
| TOP [29] | 1.71 M | $40.50 \pm 4.75$ | $8.91 \pm 0.31$ |
| ViLa-MIL [34] | 2.32 M | $19.03 \pm 2.07$ | $5.42 \pm 0.20$ |
| MSCPT [11] | 1.35 M | $41.86 \pm 1.00$ | $6.47 \pm 0.40$ |
| FOCUS [10] | 1.32 M | $132.41 \pm 2.26$ | $30.11 \pm 1.93$ |
| MAPLE | 1.86 M | $45.88 \pm 1.18$ | $10.52 \pm 0.19$ |

## B.3 Computational Complexity

We analyze the computational efficiency of MAPLE in comparison with existing few-shot methods. As shown in Table 4, we report the number of trainable parameters, inference time per slide, and training time per epoch on the TCGA-NSCLC dataset. Among all methods, ViLa-MIL requires the largest number of trainable parameters (2.32M) due to its dual-scale visual prompt tuning approach, though it achieves the fastest inference and training speeds. In contrast, FOCUS has a relatively small parameter count (1.32M) but incurs the highest computational cost during both inference (132.41 ms/slide) and training (30.11 s/epoch). MSCPT maintains low parameter count and reasonable efficiency but requires an additional preprocessing step to select low-resolution patches, which is not reflected in the runtime measurements. MAPLE strikes a balanced trade-off between model complexity and computational efficiency. While it requires moderately more parameters (1.86M) than TOP, MSCPT, and FOCUS, its inference and training times remain comparable to TOP and significantly lower than FOCUS. Overall, MAPLE achieves strong performance in few-shot scenarios without introducing significant additional trainable parameters or increasing much inference and training time.

## C   Additional Results

### C.1   Comparisons with SOTAs using CLIP

In addition to the main experiments using PLIP [14] as the feature extractor, we conducted parallel experiments using CLIP [30] to extract visual and textual features. Tab. 5 presents the few-shot weakly-supervised learning results on the same three datasets (TCGA-BRCA, TCGA-RCC, and TCGA-NSCLC) across 4-shot, 8-shot, and 16-shot settings. With features extracted by the CLIP encoder, MAPLE achieves the best results in nearly all configurations across the three datasets. Particularly, in the 16-shot setting, MAPLE attains the highest AUC scores of 70.1%, 94.6%, and 76.0% on TCGA-BRCA, TCGA-RCC, and TCGA-NSCLC, respectively. The performance advantage of MAPLE is maintained even in the challenging 4-shot scenario, where it achieves competitive or superior results compared to other methods.

**Table 5:** Few-shot WSI classification results using CLIP encoder on TCGA-BRCA, TCGA-RCC, and TCGA-NSCLC datasets under 4-shot, 8-shot, and 16-shot settings. The best results are in **bold**, and the second-best results are underlined.

| Dataset | Methods | TCGA-BRCA | | | TCGA-RCC | | | TCGA-NSCLC | | |
|---|---|---|---|---|---|---|---|---|---|---|
| | | AUC | F1 | ACC | AUC | F1 | ACC | AUC | F1 | ACC |
| 4-shot | ABMIL | 0.549 ± 0.083 | 0.428 ± 0.104 | 0.526 ± 0.173 | 0.825 ± 0.033 | 0.586 ± 0.077 | 0.597 ± 0.086 | 0.589 ± 0.052 | 0.545 ± 0.042 | 0.557 ± 0.037 |
| | TransMIL | 0.560 ± 0.058 | 0.396 ± 0.107 | 0.466 ± 0.161 | 0.759 ± 0.052 | 0.482 ± 0.081 | 0.518 ± 0.080 | 0.547 ± 0.064 | 0.519 ± 0.048 | 0.531 ± 0.052 |
| | GTMIL | 0.570 ± 0.108 | 0.446 ± 0.126 | 0.456 ± 0.154 | 0.816 ± 0.046 | 0.598 ± 0.090 | 0.606 ± 0.115 | 0.579 ± 0.044 | 0.476 ± 0.044 | 0.531 ± 0.027 |
| | WiKG | 0.581 ± 0.089 | 0.495 ± 0.036 | 0.615 ± 0.070 | 0.820 ± 0.048 | 0.572 ± 0.099 | 0.592 ± 0.121 | 0.600 ± 0.063 | 0.567 ± 0.054 | 0.576 ± 0.055 |
| | TOP | 0.573 ± 0.053 | 0.480 ± 0.025 | 0.588 ± 0.110 | 0.816 ± 0.032 | 0.534 ± 0.117 | 0.576 ± 0.121 | 0.627 ± 0.029 | 0.490 ± 0.111 | 0.557 ± 0.047 |
| | ViLa-MIL | **0.614 ± 0.066** | 0.499 ± 0.082 | 0.560 ± 0.120 | 0.806 ± 0.033 | 0.525 ± 0.129 | 0.565 ± 0.115 | 0.665 ± 0.080 | 0.585 ± 0.041 | 0.606 ± 0.046 |
| | MSCPT | 0.610 ± 0.071 | 0.476 ± 0.090 | 0.544 ± 0.133 | 0.828 ± 0.027 | 0.608 ± 0.078 | 0.613 ± 0.079 | 0.581 ± 0.032 | 0.495 ± 0.062 | 0.531 ± 0.021 |
| | FOCUS | 0.602 ± 0.067 | 0.483 ± 0.046 | 0.600 ± 0.133 | 0.823 ± 0.064 | 0.619 ± 0.105 | 0.618 ± 0.109 | 0.574 ± 0.069 | 0.511 ± 0.111 | 0.548 ± 0.064 |
| | **MAPLE** | 0.613 ± 0.100 | **0.525 ± 0.043** | **0.637 ± 0.091** | **0.831 ± 0.038** | **0.635 ± 0.080** | **0.631 ± 0.086** | **0.678 ± 0.057** | **0.589 ± 0.086** | **0.616 ± 0.051** |
| 8-shot | ABMIL | 0.577 ± 0.079 | 0.435 ± 0.111 | 0.534 ± 0.198 | 0.830 ± 0.023 | 0.618 ± 0.023 | 0.664 ± 0.048 | 0.590 ± 0.079 | 0.548 ± 0.059 | 0.561 ± 0.059 |
| | TransMIL | 0.553 ± 0.051 | 0.451 ± 0.071 | 0.504 ± 0.135 | 0.777 ± 0.064 | 0.571 ± 0.068 | 0.596 ± 0.071 | 0.559 ± 0.061 | 0.474 ± 0.078 | 0.526 ± 0.526 |
| | GTMIL | 0.594 ± 0.063 | 0.496 ± 0.024 | 0.565 ± 0.045 | 0.877 ± 0.013 | 0.668 ± 0.022 | 0.684 ± 0.020 | 0.629 ± 0.100 | 0.550 ± 0.050 | 0.563 ± 0.051 |
| | WiKG | 0.652 ± 0.063 | 0.504 ± 0.080 | 0.542 ± 0.119 | 0.874 ± 0.020 | 0.671 ± 0.043 | 0.690 ± 0.046 | 0.609 ± 0.096 | 0.545 ± 0.053 | 0.559 ± 0.056 |
| | TOP | 0.632 ± 0.067 | 0.536 ± 0.115 | 0.548 ± 0.113 | 0.786 ± 0.034 | 0.580 ± 0.065 | 0.625 ± 0.059 | 0.591 ± 0.020 | 0.401 ± 0.078 | 0.513 ± 0.026 |
| | ViLa-MIL | 0.645 ± 0.099 | 0.537 ± 0.134 | 0.558 ± 0.174 | 0.865 ± 0.022 | 0.685 ± 0.041 | 0.714 ± 0.041 | 0.652 ± 0.036 | 0.606 ± 0.032 | 0.615 ± 0.024 |
| | MSCPT | 0.629 ± 0.077 | 0.524 ± 0.100 | 0.562 ± 0.182 | 0.866 ± 0.035 | 0.673 ± 0.050 | 0.705 ± 0.048 | 0.548 ± 0.034 | 0.507 ± 0.072 | 0.531 ± 0.039 |
| | FOCUS | 0.641 ± 0.074 | 0.541 ± 0.064 | 0.605 ± 0.102 | 0.888 ± 0.029 | 0.695 ± 0.037 | 0.719 ± 0.044 | 0.599 ± 0.088 | 0.567 ± 0.072 | 0.575 ± 0.064 |
| | **MAPLE** | **0.658 ± 0.072** | **0.566 ± 0.045** | **0.631 ± 0.052** | 0.886 ± 0.026 | 0.689 ± 0.038 | 0.708 ± 0.048 | **0.668 ± 0.043** | **0.612 ± 0.055** | **0.620 ± 0.042** |
| 16-shot | ABMIL | 0.629 ± 0.082 | 0.536 ± 0.059 | 0.590 ± 0.074 | 0.900 ± 0.018 | 0.714 ± 0.041 | 0.728 ± 0.042 | 0.687 ± 0.044 | 0.634 ± 0.052 | 0.638 ± 0.049 |
| | TransMIL | 0.590 ± 0.031 | 0.464 ± 0.114 | 0.537 ± 0.169 | 0.860 ± 0.029 | 0.587 ± 0.127 | 0.614 ± 0.115 | 0.684 ± 0.016 | 0.599 ± 0.033 | 0.618 ± 0.027 |
| | GTMIL | 0.652 ± 0.082 | 0.532 ± 0.053 | 0.593 ± 0.092 | 0.917 ± 0.026 | 0.757 ± 0.045 | 0.766 ± 0.054 | 0.693 ± 0.037 | 0.632 ± 0.036 | 0.635 ± 0.036 |
| | WiKG | 0.686 ± 0.067 | 0.515 ± 0.098 | 0.558 ± 0.130 | 0.926 ± 0.010 | 0.757 ± 0.041 | 0.777 ± 0.034 | 0.730 ± 0.042 | 0.680 ± 0.031 | 0.684 ± 0.032 |
| | TOP | 0.630 ± 0.053 | 0.483 ± 0.102 | 0.505 ± 0.160 | 0.914 ± 0.020 | 0.745 ± 0.024 | 0.760 ± 0.042 | 0.672 ± 0.063 | 0.618 ± 0.046 | 0.624 ± 0.043 |
| | ViLa-MIL | 0.690 ± 0.047 | 0.533 ± 0.083 | 0.582 ± 0.106 | 0.937 ± 0.009 | 0.779 ± 0.026 | 0.774 ± 0.024 | 0.744 ± 0.057 | 0.674 ± 0.062 | 0.682 ± 0.056 |
| | MSCPT | 0.669 ± 0.073 | 0.540 ± 0.050 | 0.601 ± 0.092 | 0.925 ± 0.010 | 0.755 ± 0.048 | 0.773 ± 0.048 | 0.732 ± 0.056 | 0.658 ± 0.043 | 0.675 ± 0.044 |
| | FOCUS | 0.662 ± 0.073 | 0.528 ± 0.065 | 0.575 ± 0.076 | 0.939 ± 0.012 | 0.789 ± 0.020 | 0.788 ± 0.020 | 0.732 ± 0.050 | 0.666 ± 0.050 | 0.669 ± 0.048 |
| | **MAPLE** | **0.701 ± 0.062** | **0.583 ± 0.049** | **0.689 ± 0.097** | **0.946 ± 0.009** | **0.804 ± 0.025** | **0.807 ± 0.027** | **0.760 ± 0.072** | **0.687 ± 0.068** | **0.693 ± 0.065** |

## C.2 Comparisons between PLIP and CLIP

Comparing the results obtained with CLIP (Tab. 5) and PLIP (Tab. 1 in the main paper), we observe that PLIP generally provides better feature representations for pathology images, resulting in higher overall performance across all metrics and settings. This observation is consistent with previous studies [14, 11] suggesting that PLIP, which is pre-trained specifically on pathology images, captures more relevant pathological features than the general-purpose CLIP model.

## C.3 Comparisons with SOTAs using CONCH

We provide further results with stronger pathology VLMs such as CONCH on the three datasets (TCGA-BRCA, TCGA-RCC and TCGA-NSCLC) in Tab. 6. MAPLE consistently outperforms baseline methods across different datasets and few-shot settings. Specifically, in the 16-shot setting, MAPLE achieves the highest AUC scores of 91.6%, 98.4%, and 98.1% on TCGA-BRCA, TCGA-RCC, and TCGA-NSCLC, respectively. In the challenging 4-shot setting, MAPLE maintains its superior performance with AUC scores of 84.4%, 94.7% and 88.9%. In summary, MAPLE can achieve consistently superior prediction results under different vision-language foundation models (e.g., CLIP, PLIP, CONCH), highlighting the advantages of our MAPLE to jointly integrate multi-scale visual semantics and perform prediction at both the entity and slide levels.

## C.4 Comparisons with multi-scale MIL methods

We further compare MAPLE with representative multi-scale MIL baselines, including DTFD-MIL [40], Dual-Stream-MIL [18] and Cross-Scale MIL [8]. As shown in Table 7, MAPLE consistently surpasses these approaches under all the settings across different datasets, highlighting the advantage of our multi-scale prompt-guided VLM-based method. These results further confirm that the performance gain of MAPLE arises not merely from multi-scale integration, but from the combination of multi-scale modeling and language-guide prompt supervision via vision-language models.

**Table 6:** Few-shot WSI classification results using CONCH encoder on TCGA-BRCA, TCGA-RCC, and TCGA-NSCLC datasets under 4-shot, 8-shot, and 16-shot settings. The best results are in **bold**, and the second-best results are underlined.

| Dataset | Methods | TCGA-BRCA | | | TCGA-RCC | | | TCGA-NSCLC | | |
|---|---|---|---|---|---|---|---|---|---|---|
| | | AUC | F1 | ACC | AUC | F1 | ACC | AUC | F1 | ACC |
| 4-shot | ABMIL | 0.770 ± 0.069 | 0.591 ± 0.070 | 0.648 ± 0.105 | 0.924 ± 0.026 | 0.755 ± 0.063 | 0.773 ± 0.062 | 0.832 ± 0.041 | 0.714 ± 0.049 | 0.721 ± 0.044 |
| | TransMIL | 0.757 ± 0.127 | 0.602 ± 0.111 | 0.672 ± 0.126 | 0.938 ± 0.019 | 0.761 ± 0.098 | 0.778 ± 0.099 | 0.848 ± 0.059 | 0.756 ± 0.125 | 0.762 ± 0.109 |
| | GTMIL | 0.728 ± 0.102 | 0.552 ± 0.094 | 0.630 ± 0.141 | 0.923 ± 0.031 | 0.743 ± 0.019 | 0.765 ± 0.054 | 0.836 ± 0.093 | 0.750 ± 0.080 | 0.754 ± 0.081 |
| | WiKG | 0.768 ± 0.109 | 0.605 ± 0.105 | 0.651 ± 0.123 | 0.925 ± 0.010 | 0.761 ± 0.038 | 0.777 ± 0.036 | 0.820 ± 0.087 | 0.730 ± 0.084 | 0.733 ± 0.083 |
| | TOP | 0.728 ± 0.170 | 0.583 ± 0.117 | 0.638 ± 0.105 | 0.916 ± 0.033 | 0.743 ± 0.057 | 0.758 ± 0.053 | 0.816 ± 0.066 | 0.683 ± 0.130 | 0.707 ± 0.092 |
| | ViLa-MIL | 0.783 ± 0.108 | 0.590 ± 0.110 | 0.635 ± 0.130 | 0.919 ± 0.030 | 0.768 ± 0.058 | 0.793 ± 0.051 | 0.853 ± 0.073 | 0.759 ± 0.081 | 0.756 ± 0.082 |
| | MSCPT | 0.782 ± 0.087 | 0.605 ± 0.072 | 0.632 ± 0.088 | 0.931 ± 0.019 | 0.770 ± 0.057 | 0.785 ± 0.050 | 0.842 ± 0.059 | 0.730 ± 0.068 | 0.748 ± 0.066 |
| | FOCUS | 0.810 ± 0.115 | 0.632 ± 0.135 | 0.667 ± 0.160 | 0.930 ± 0.032 | 0.767 ± 0.075 | 0.780 ± 0.065 | 0.875 ± 0.077 | 0.762 ± 0.060 | 0.769 ± 0.061 |
| | **MAPLE** | **0.844 ± 0.109** | **0.653 ± 0.126** | **0.695 ± 0.166** | **0.947 ± 0.017** | **0.791 ± 0.086** | **0.805 ± 0.069** | **0.889 ± 0.055** | **0.774 ± 0.032** | **0.786 ± 0.033** |
| 8-shot | ABMIL | 0.857 ± 0.049 | 0.705 ± 0.057 | 0.763 ± 0.058 | 0.941 ± 0.013 | 0.835 ± 0.055 | 0.844 ± 0.047 | 0.925 ± 0.013 | 0.835 ± 0.021 | 0.835 ± 0.021 |
| | TransMIL | 0.853 ± 0.044 | 0.710 ± 0.044 | 0.771 ± 0.046 | 0.940 ± 0.012 | 0.840 ± 0.033 | 0.851 ± 0.024 | 0.916 ± 0.010 | 0.821 ± 0.027 | 0.821 ± 0.027 |
| | GTMIL | 0.861 ± 0.052 | 0.711 ± 0.068 | 0.780 ± 0.065 | 0.945 ± 0.006 | 0.845 ± 0.020 | 0.857 ± 0.016 | 0.928 ± 0.016 | 0.844 ± 0.020 | 0.844 ± 0.020 |
| | WiKG | 0.851 ± 0.045 | 0.685 ± 0.050 | 0.741 ± 0.060 | 0.947 ± 0.012 | 0.834 ± 0.025 | 0.855 ± 0.020 | 0.919 ± 0.007 | 0.834 ± 0.021 | 0.834 ± 0.020 |
| | TOP | 0.859 ± 0.028 | 0.701 ± 0.050 | 0.758 ± 0.054 | 0.929 ± 0.020 | 0.814 ± 0.039 | 0.828 ± 0.035 | 0.908 ± 0.043 | 0.817 ± 0.068 | 0.818 ± 0.067 |
| | ViLa-MIL | 0.880 ± 0.081 | 0.726 ± 0.117 | 0.776 ± 0.090 | 0.945 ± 0.008 | 0.835 ± 0.037 | 0.858 ± 0.032 | 0.934 ± 0.037 | 0.856 ± 0.051 | 0.857 ± 0.051 |
| | MSCPT | 0.882 ± 0.091 | 0.720 ± 0.132 | 0.774 ± 0.136 | 0.950 ± 0.011 | 0.849 ± 0.043 | 0.851 ± 0.041 | 0.924 ± 0.043 | 0.849 ± 0.055 | 0.849 ± 0.055 |
| | FOCUS | 0.875 ± 0.060 | 0.719 ± 0.114 | 0.747 ± 0.145 | 0.959 ± 0.008 | 0.871 ± 0.033 | 0.875 ± 0.031 | 0.949 ± 0.030 | 0.873 ± 0.043 | 0.873 ± 0.042 |
| | **MAPLE** | **0.900 ± 0.082** | **0.748 ± 0.110** | **0.797 ± 0.120** | **0.971 ± 0.011** | **0.888 ± 0.025** | **0.899 ± 0.023** | **0.964 ± 0.032** | **0.894 ± 0.033** | **0.894 ± 0.034** |
| 16-shot | ABMIL | 0.876 ± 0.017 | 0.759 ± 0.020 | 0.789 ± 0.018 | 0.954 ± 0.004 | 0.857 ± 0.009 | 0.859 ± 0.007 | 0.935 ± 0.013 | 0.865 ± 0.021 | 0.865 ± 0.021 |
| | TransMIL | 0.884 ± 0.030 | 0.761 ± 0.057 | 0.795 ± 0.057 | 0.955 ± 0.003 | 0.854 ± 0.010 | 0.860 ± 0.013 | 0.926 ± 0.014 | 0.851 ± 0.027 | 0.851 ± 0.027 |
| | GTMIL | 0.891 ± 0.025 | 0.770 ± 0.073 | 0.803 ± 0.083 | 0.962 ± 0.004 | 0.865 ± 0.008 | 0.878 ± 0.013 | 0.938 ± 0.016 | 0.874 ± 0.020 | 0.874 ± 0.020 |
| | WiKG | 0.882 ± 0.013 | 0.762 ± 0.019 | 0.796 ± 0.018 | 0.957 ± 0.010 | 0.839 ± 0.031 | 0.856 ± 0.031 | 0.939 ± 0.007 | 0.864 ± 0.021 | 0.864 ± 0.020 |
| | TOP | 0.887 ± 0.011 | 0.768 ± 0.016 | 0.790 ± 0.015 | 0.944 ± 0.003 | 0.829 ± 0.027 | 0.835 ± 0.027 | 0.924 ± 0.013 | 0.859 ± 0.025 | 0.859 ± 0.025 |
| | ViLa-MIL | 0.902 ± 0.033 | 0.775 ± 0.038 | 0.812 ± 0.042 | 0.966 ± 0.006 | 0.862 ± 0.024 | 0.872 ± 0.016 | 0.941 ± 0.023 | 0.877 ± 0.028 | 0.877 ± 0.017 |
| | MSCPT | 0.894 ± 0.018 | 0.767 ± 0.027 | 0.808 ± 0.024 | 0.958 ± 0.004 | 0.859 ± 0.021 | 0.871 ± 0.025 | 0.934 ± 0.017 | 0.866 ± 0.031 | 0.867 ± 0.031 |
| | FOCUS | 0.893 ± 0.017 | 0.764 ± 0.041 | 0.805 ± 0.042 | 0.974 ± 0.006 | 0.884 ± 0.045 | 0.891 ± 0.051 | 0.964 ± 0.007 | 0.894 ± 0.052 | 0.895 ± 0.050 |
| | **MAPLE** | **0.916 ± 0.024** | **0.790 ± 0.029** | **0.825 ± 0.030** | **0.984 ± 0.006** | **0.910 ± 0.016** | **0.919 ± 0.015** | **0.981 ± 0.005** | **0.914 ± 0.050** | **0.914 ± 0.047** |

**Table 7:** Comparisons of few-shot WSI classification results using multi-scale MIL methods on TCGA-BRCA, TCGA-RCC, and TCGA-NSCLC datasets under 4-shot, 8-shot, and 16-shot settings.

| Dataset | Methods | TCGA-BRCA | | | TCGA-RCC | | | TCGA-NSCLC | | |
|---|---|---|---|---|---|---|---|---|---|---|
| | | AUC | F1 | ACC | AUC | F1 | ACC | AUC | F1 | ACC |
| 4-shot | DTFD-MIL | 0.648 ± 0.050 | 0.520 ± 0.065 | 0.593 ± 0.068 | 0.869 ± 0.031 | 0.626 ± 0.057 | 0.660 ± 0.070 | 0.624 ± 0.049 | 0.555 ± 0.051 | 0.581 ± 0.046 |
| | Dual-Stream MIL | 0.669 ± 0.052 | 0.562 ± 0.132 | 0.623 ± 0.162 | 0.877 ± 0.039 | 0.664 ± 0.037 | 0.693 ± 0.036 | 0.649 ± 0.122 | 0.564 ± 0.094 | 0.590 ± 0.080 |
| | Cross-Scale MIL | 0.673 ± 0.097 | 0.552 ± 0.061 | 0.615 ± 0.068 | 0.876 ± 0.019 | 0.667 ± 0.037 | 0.690 ± 0.039 | 0.560 ± 0.093 | 0.560 ± 0.093 | 0.586 ± 0.084 |
| | **MAPLE** | **0.722 ± 0.063** | **0.594 ± 0.076** | **0.664 ± 0.134** | **0.909 ± 0.020** | **0.705 ± 0.055** | **0.728 ± 0.057** | **0.740 ± 0.056** | **0.663 ± 0.052** | **0.675 ± 0.053** |
| 8-shot | DTFD-MIL | 0.733 ± 0.048 | 0.546 ± 0.061 | 0.603 ± 0.086 | 0.907 ± 0.024 | 0.722 ± 0.036 | 0.748 ± 0.046 | 0.729 ± 0.040 | 0.632 ± 0.042 | 0.652 ± 0.030 |
| | Dual-Stream MIL | 0.758 ± 0.073 | 0.548 ± 0.071 | 0.576 ± 0.088 | 0.926 ± 0.025 | 0.761 ± 0.041 | 0.789 ± 0.037 | 0.752 ± 0.074 | 0.651 ± 0.032 | 0.667 ± 0.034 |
| | Cross-Scale MIL | 0.756 ± 0.062 | 0.554 ± 0.063 | 0.588 ± 0.075 | 0.924 ± 0.023 | 0.757 ± 0.028 | 0.782 ± 0.031 | 0.748 ± 0.037 | 0.645 ± 0.075 | 0.657 ± 0.071 |
| | **MAPLE** | **0.786 ± 0.070** | **0.618 ± 0.024** | **0.673 ± 0.018** | **0.957 ± 0.015** | **0.791 ± 0.024** | **0.806 ± 0.024** | **0.855 ± 0.041** | **0.762 ± 0.031** | **0.766 ± 0.030** |
| 16-shot | DTFD-MIL | 0.738 ± 0.044 | 0.623 ± 0.064 | 0.679 ± 0.088 | 0.919 ± 0.019 | 0.762 ± 0.051 | 0.799 ± 0.040 | 0.812 ± 0.047 | 0.742 ± 0.044 | 0.747 ± 0.042 |
| | Dual-Stream MIL | 0.752 ± 0.038 | 0.636 ± 0.059 | 0.696 ± 0.062 | 0.946 ± 0.014 | 0.813 ± 0.030 | 0.827 ± 0.032 | 0.824 ± 0.029 | 0.760 ± 0.037 | 0.762 ± 0.032 |
| | Cross-Scale MIL | 0.759 ± 0.052 | 0.632 ± 0.055 | 0.698 ± 0.060 | 0.948 ± 0.019 | 0.815 ± 0.034 | 0.830 ± 0.035 | 0.830 ± 0.042 | 0.764 ± 0.029 | 0.768 ± 0.028 |
| | **MAPLE** | **0.801 ± 0.031** | **0.672 ± 0.076** | **0.735 ± 0.039** | **0.969 ± 0.014** | **0.838 ± 0.034** | **0.867 ± 0.031** | **0.903 ± 0.033** | **0.806 ± 0.060** | **0.810 ± 0.055** |

# D    More Ablation Studies

## D.1    Number of Neighbors

To analyze the effect of the number of neighbors $k$ used in the cross-scale entity graph, we set $n_k \in \{1, 3, 5, 7, 9, 11, 13\}$. As shown in Fig. 7, the performance improves as $n_k$ increases, and achieves the best result at $n_k$ of 7. It demonstrates the importance of contextual information propagation between semantically related entities. However, further increasing $n_k$ leads to slight degradation, likely due to over-smoothing caused by excessive message propagation across weakly related entities.

## D.2    Number of Entities

We further investigate how the number of selected entities per scale affects the results. As illustrated in Fig. 8, increasing the number of entities from 4 to 8 improves the performance, indicating the importance of providing enough entities to capture different fine-grained subtype-specific patterns. Beyond 8 entities, performance plateaus and even slightly decreases, suggesting that additional entities may lack discriminative phenotypic attributes relevant to cancer subtype classification. These

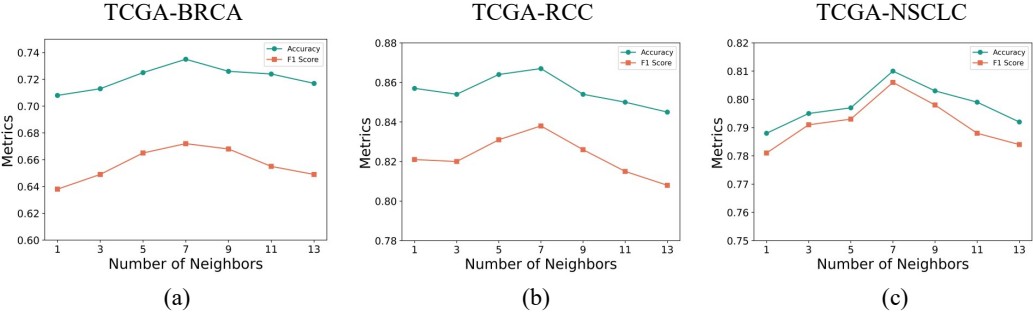

**Figure 7:** Impact of the number of neighbor nodes across three datasets under the 16-shot setting.

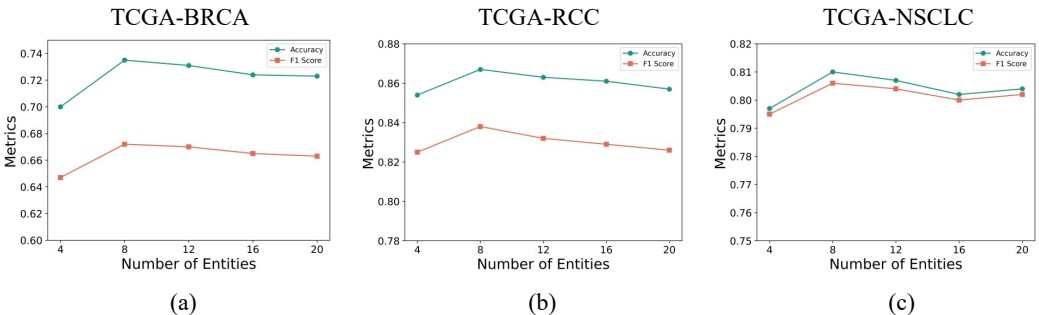

**Figure 8:** Impact of the number of entities across three datasets under the 16-shot setting.

redundant entities do not contribute to improving the model's performance but increase computational complexity and potentially introduce noise that interferes with the model's decision-making process.

### D.3 Number of Tumor-related Patches

To assess the effects of the number of tumor-related patches, we vary $r$ from 0.1 to 0.9. As shown in Fig. 9, performance improves steadily as $r$ increases from 0.1 to 0.7, demonstrating the importance of incorporating sufficient tumor-related patches for accurate diagnosis. Further increasing $r$ yields the slight performance degradation. This decline can be attributed to the inclusion of less informative or non-tumor regions that compromise the quality of the selected patch set, introducing noise that interferes with entity-level feature extraction.

### D.4 Impact of Lambda

To balance contributions from entity-level and slide-level classification, we vary the fusion weight $\lambda$ from 0 to 1 in increments of 0.1 (Fig. 10). Results indicate that $\lambda = 0.3$ offers the best trade-off, confirming the importance of leveraging both entity-level and slide-level information for few-shot WSI classification.

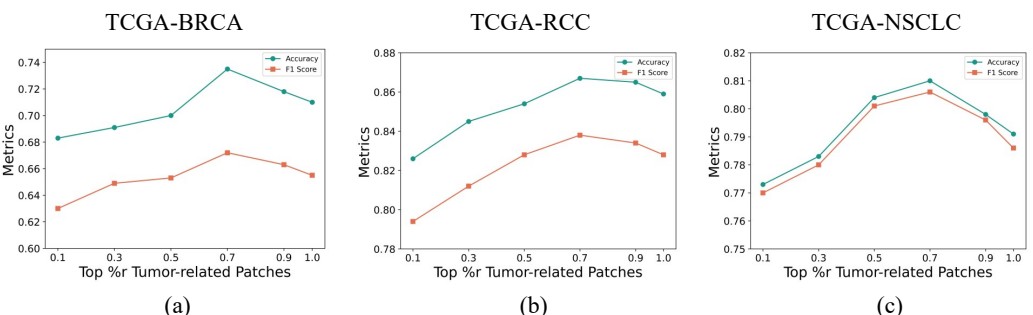

**Figure 9:** Impact of the number of tumor-related patches across three datasets under the 16-shot setting.

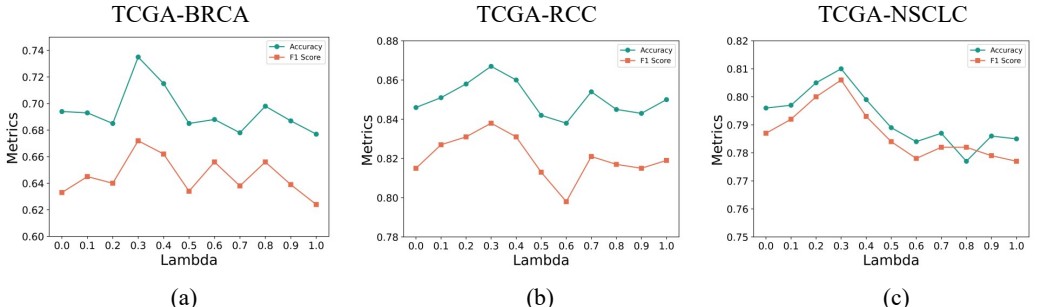

**Figure 10:** Impact of $\lambda$ across three datasets under the 16-shot setting.

**Table 8:** Results of different large language models on the three datasets under the 16-shot setting.

| LLMs | TCGA-BRCA | | | TCGA-RCC | | | TCGA-NSCLC | | |
|---|---|---|---|---|---|---|---|---|---|
| | AUC | F1 | ACC | AUC | F1 | ACC | AUC | F1 | ACC |
| Claude 3.5 Sonnet | $0.786 \pm 0.031$ | $0.647 \pm 0.083$ | $0.719 \pm 0.107$ | $0.966 \pm 0.018$ | $0.822 \pm 0.072$ | $0.854 \pm 0.073$ | $0.886 \pm 0.039$ | $0.791 \pm 0.013$ | $0.795 \pm 0.013$ |
| Qwen2.5 [38] | $0.795 \pm 0.041$ | $0.664 \pm 0.031$ | $0.730 \pm 0.054$ | $0.961 \pm 0.017$ | $0.811 \pm 0.055$ | $0.845 \pm 0.055$ | $0.885 \pm 0.018$ | $0.790 \pm 0.021$ | $0.794 \pm 0.021$ |
| Deepseek-V3 [22] | $0.799 \pm 0.027$ | $0.665 \pm 0.019$ | $0.732 \pm 0.045$ | $0.968 \pm 0.013$ | $0.833 \pm 0.038$ | $0.863 \pm 0.037$ | $\mathbf{0.903 \pm 0.027}$ | $\mathbf{0.813 \pm 0.012}$ | $\mathbf{0.817 \pm 0.012}$ |
| GPT-4 [2] | $\mathbf{0.801 \pm 0.031}$ | $\mathbf{0.672 \pm 0.076}$ | $\mathbf{0.735 \pm 0.039}$ | $\mathbf{0.969 \pm 0.014}$ | $\mathbf{0.838 \pm 0.034}$ | $\mathbf{0.867 \pm 0.031}$ | $\mathbf{0.903 \pm 0.033}$ | $0.806 \pm 0.060$ | $0.810 \pm 0.055$ |

## D.5 Impact of Large Language Models

To investigate how the choice of large language model affects performance of MAPLE, we evaluate four LLMs for prompt construction: Claude 3.5 Sonnet, Qwen2.5 [38], Deepseek-V3 [22], and GPT-4 [2]. For each LLM, we use it to generate both entity-level and slide-level prompts following the same query templates. We present the results on the three datasets under 16-shot setting in Tab. 8. As shown in Tab. 8, all four LLMs deliver strong performance across the three datasets, indicating the robustness of our method to LLMs. GPT-4 achieves the best overall results, particularly on TCGA-BRCA and TCGA-RCC datasets, while Deepseek-V3 performs comparably and even slightly outperforms GPT-4 on TCGA-NSCLC in terms of F1 score and accuracy. Qwen2.5 and Claude 3.5 Sonnet also demonstrate competitive performance, with marginally lower metrics compared to GPT-4 and Deepseek-V3. The findings also indicate that LLMs with stronger language modeling capabilities can further enhance performance of MAPLE.

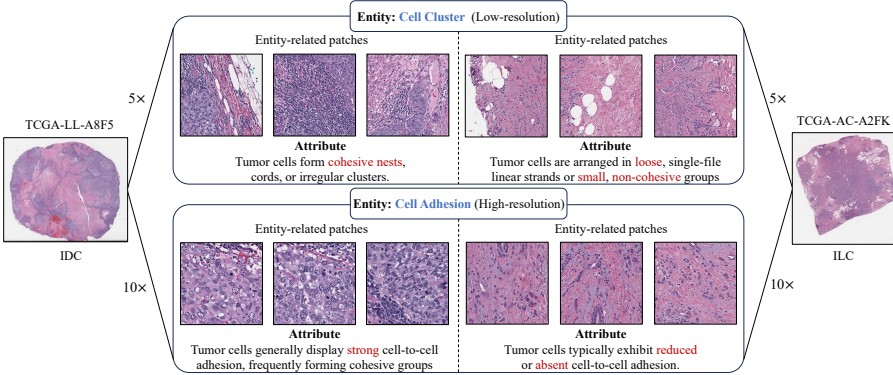

**Figure 11:** Visualization of entity-relevant patches selected by the entity-guided cross-attention module for invasive ductal carcinoma (IDC) and invasive lobular carcinoma (ILC) on the TCGA-BRCA dataset. Top rows show patches and their corresponding entity attributes (*e.g.,* cell cluster) at low resolution, while bottom rows show patches and their corresponding entity attributes (*e.g.,* cell adhesion) at high resolution.

## E More Visualization Results

In this section, we provide more visualization results of entities and their relevant patches on the TCGA-BRCA and TCGA-RCC in Fig. 11 and 12, respectively.

As shown in Fig. 11, at low resolution, cell cluster-related patches from invasive ductal carcinoma (IDC) consistently form cohesive nests, while those from invasive lobular carcinoma (ILC) are arranged in small, non-cohesive groups. Similarly, at high resolution, cell adhesion-related patches

from IDC display strong cell-to-cell adhesion, whereas ILC patches exhibit reduced or absent intercellular adhesion. These visualizations align with the subtype-specific attributes of these entities described in our LLM-generated prompts.

We observe similarly patterns in the multi-class TCGA-RCC dataset. As illustrated in Fig. 12, at low resolution, stroma-related patches exhibit clear subtype-specific characteristics: clear cell renal cell carcinoma (CCRCC) shows delicate and inconspicuous stroma with highly vascular background; chromophobe renal cell carcinoma (CHRCC) displays dense, hyalinized fibrous stroma with less prominent blood vessels; and papillary renal cell carcinoma (PRCC) features prominent fibrovascular cores supporting papillary fronds with abundant stroma. At high resolution, vacuole-related patches from CCRCC displays prominent, large vacuoles; patches from CHRCC exhibits multiple small, well-defined cytoplasmic vacuoles; while patches from PRCC shows small, inconspicuous vacuoles that are typically less pronounced. These visualizations confirm that MAPLE effectively captures the histological entities and their attributes for subtype classification.

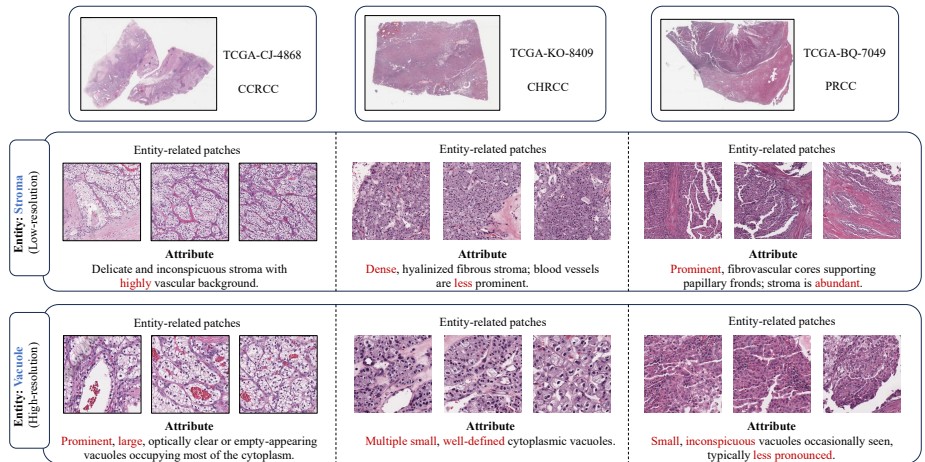

**Figure 12:** Visualization of entity-relevant patches selected by the entity-guided cross-attention module for clear cell renal cell carcinoma (CCRCC), chromophobe renal cell carcinoma (CHRCC) and papillary renal cell carcinoma (PRCC) on the TCGA-RCC dataset. Top rows show patches and their corresponding entity attributes (*e.g.,* stroma) at low resolution, while bottom rows show patches and their corresponding entity attributes (*e.g.,* vacuole) at high resolution.

# F    Discussion

## F.1    Limitations

The construction of entity and attribute prompts currently relies solely on LLMs such as GPT-4. Although LLMs provide rich semantic priors, they may introduce hallucinations or generate clinically irrelevant descriptions. This may limit the reliability of the derived prompts in real-world settings. In future work, we aim to incorporate domain expertise from pathologists into the prompt design process. This could involve human-in-the-loop strategies for verifying or refining entity-attribute relationships, or integrating expert-annotated diagnostic criteria to guide prompt construction.

## F.2    Broader Impacts

MAPLE has the potential to significantly impact clinical pathology practice and cancer diagnosis workflows. By providing accurate few-shot WSI classification with interpretable entity-level predictions, our method could help address critical challenges in computational pathology: First, MAPLE could reduce the annotation burden for pathologists by enabling accurate diagnosis with limited labeled examples, particularly valuable for rare cancer subtypes where collecting large labeled datasets is challenging. Additionally, the hierarchical framework may also enhance collaboration between AI systems and pathologists by providing entity-level interpretations that align with human diagnostic reasoning.

