# OpenReview forum: "MAPLE: Multi-scale Attribute-enhanced Prompt Learning for Few-shot Whole Slide Image Classification"
_NeurIPS.cc/2025/Conference — NeurIPS 2025 poster_

### Official Review · Reviewer_yzxQ · 2025-06-23

**Clarity:** 3
**Significance:** 2
**Originality:** 3
**Rating:** 4
**Confidence:** 5

**Summary:**

This work leverages large language models (LLMs) to generate both entity-level and slide-level prompts, integrating fine-grained and coarse-grained visual and textual information for few-shot whole-slide image classification. The proposed MAPLE method consistently outperforms existing SOTA models in this field. The integration of entity-level knowledge is promising to boost few-shot classification performance, especially in the era of LLMs.

**Questions:**

1. Reliability of using LLMs to generate knowledge-required descriptions for entities. Although it’s common practice to utilize LLMs for prompt or diagnosis description generation, it would be better to demonstrate the reliability of the generated prompts. For example, do they accurately describe the slides’ morphology of a certain cancer? I see that Table 6 provides an ablation of the impact of using different LLMs for prompt generation. One thing I’d like to discuss further is that existing works mostly conduct this kind of ablation, yet sometimes the results are not consistent due to different environments or settings. For example, in FOCUS, they show that Claude demonstrates the strongest performance for prompt generation. This kind of difference leads to difficulty in choosing appropriate LLMs in clinical scenarios. Could the authors share some insights regarding this problem?
2. As mentioned in the Weaknesses part, could the authors provide more explanation regarding the choice of using PLIP as the feature extractor (although there is a comparison between PLIP and CLIP, CLIP achieves even worse performance)? There have been many powerful pathology foundation models (or feature extractors) in recent years, such as UNI [1], GPFM [2], and Virchow [3]. A major concern here is that existing models might use different feature encoders to extract features, for example, FOCUS uses CONCH [4] as the visual encoder. Thus, this lack of sufficient ablation study on feature encoders could lead to unfair comparisons.
3. Meanwhile, from Table 1, it could be seen that the results are relatively low compared to ones reported from other works. Still take FOCUS for example, the 16-shot classification results of TCGA-NSCLC are around 0.95 AUC for several models. Yet in this work, the same task under the same few-shot setting only achieves around 0.9 AUC. I believe this also stems from the choice of feature extractor, i.e., PLIP.
4. Could the authors explain more about the visualization details in section 4.3? Are the patches shown for each entity selected from the ones with the highest attention score?
5. An additional question is that, if the entity can be precisely identified, then what’s the point of using the global description and WSI-level representation for further integration and analysis?

[1] Chen, Richard J., et al. "Towards a general-purpose foundation model for computational pathology." Nature Medicine 30.3 (2024): 850-862.

[2] Ma, Jiabo, et al. "Towards a generalizable pathology foundation model via unified knowledge distillation." arXiv preprint arXiv:2407.18449 (2024).

[3] Vorontsov, Eugene, et al. "A foundation model for clinical-grade computational pathology and rare cancers detection." Nature medicine 30.10 (2024): 2924-2935.

[4] Lu, Ming Y., et al. "A visual-language foundation model for computational pathology." Nature Medicine 30.3 (2024): 863-874.

**Ethical Concerns:**

["NO or VERY MINOR ethics concerns only"]

**Final Justification:**

After carefully reviewing the authors' response and other reviewers' comments, I believe the revised manuscript after rebuttal would satisfy NeurIPS's standards.

**Paper Formatting Concerns:**

Putting the Related Works section in the Appendix does not seem appropriate.

**Quality:**

2

**Strengths And Weaknesses:**

### Strengths:
1. This work introduces entity-level descriptions generated from LLMs to provide fine-grained information for visual-textual alignment.
2. The performance of the proposed MAPLE model is promising, surpassing existing SOTAs.
3. Experiments are relatively solid, covering three cancer cohorts from the TCGA dataset.

### Weaknesses:
1. Visuals in this work (for example, Figure 1) are not precise. For instance, the third paragraph of Introduction mentioned that “TOP [38] introduces instance-level phenotypic prompts to guide patch aggregation into slide-level features, while ViLa-MIL [43] leverages learnable visual prototypes to guide the fusion process of patch features and considers dual-scale visual descriptive text prompt to boost the performance.”, however, Figure 1(a) only illustrates the slide-level similarity computation process, which is too simplified and does not correspond to the texts. What’s more, last paragraph of Introduction mentions “MAPLE … across different scales, as illustrated in Fig. 1”, however, Fig. 1 clearly does not contain multi-scale visualization.
2. Similar concerns are in the texts as well. For example, the second paragraph of Section 2.2 describes the learning process of few-shot WSI analysis. However, it is way too simplified and only mentions the slide-level similarity computation for logits generation. Most existing works computes the logits more than this, such as ViLa-MIL [1] and FOCUS [2], which also apply patch-level similarity computation.
3. Lack of certain ablation studies. It is mentioned that PLIP is used as the feature extractor for the visual branch of MAPLE, however, as mentioned in FOCUS [2] (Figure 2 Ablation Study on Foundation Models), PLIP typically performs the worst in few-shot WSI classification. The choice of using PLIP is not well explained. Please refer to the Questions section for more related details.

[1] Shi, Jiangbo, et al. "ViLa-MIL: Dual-scale Vision-Language Multiple Instance Learning for Whole Slide Image Classification." Proceedings of the IEEE/CVF Conference on Computer Vision and Pattern Recognition. 2024.

[2] Guo, Zhengrui, et al. "Focus: Knowledge-enhanced adaptive visual compression for few-shot whole slide image classification." Proceedings of the Computer Vision and Pattern Recognition Conference. 2025.

---

> ### Author Rebuttal · Authors · 2025-07-31
>
> We thank the reviewer for the comments and for the time spent reviewing our paper. We address the weaknesses (W) and questions (Q) as follows:
>
> ---
>
> >**[W1] Clarification on Figure 1.**
>
> We thank the reviewer for pointing this out. Actually, Figure 1 is intended to emphasize the key difference in alignment strategy between our MAPLE and the existing studies (e.g., TOP and ViLa-MIL), where the existing methods only perform the slide-level alignment (shown in Fig.1(a)), while our method considers additional entity-level features and incorporates subtype-specific phenotypic attributes for more interpretable and precise alignment (shown in Fig.1(b)). Since multi-scale modeling also appears in methods like ViLa-MIL, we do not explicitly distinguish this in the figure, but note it in the caption. We will further revise the figure and related text in the final version to avoid confusion.
>
> >**[W2] Concerns on texts in Section 2.2.**
>
> We acknowledge that different methods may adopt varying strategies to compute logits, often involving additional architectural components and complex aggregation strategies. However, the purpose of Section 2.2 is to provide a concise and coherent overview of the general paradigm, rather than to detail the full implementation of all existing approaches. Therefore, we believe that our formulation captures the essential inference process commonly adopted in prompt-based few-shot WSI classification. We would be happy to further clarify any part of this formulation if needed.
>
> >**[Q1] Reliability of LLM and insights of choosing appropriate LLMs.**
>
> We acknowledge the reviewer's concern regarding the reliability of LLM-generated prompts, and we provide the detailed discussion in the response to [W3] of Reviewer dMRb. To assess the reliability of our generated prompts, we examine the entities generated from GPT4 by querying different LLMs (e.g., Claude 3.5 Sonnet, DeepSeek-V3 and Qwen2.5) using the prompt: "Do you think [ENTITY] is a key entity relevant to distinguishing different subtypes in [DATASET]?" All of these LLMs affirm the relevance and importance of the generated entities, which can demonstrate the accuracy of selected entities.
>
> For practical usage of choosing appropriate LLMs in clinical scenarios, we suggest selecting recently released LLMs that benefit from improved medical knowledge coverage and instruction-following ability. In addition, combining different outputs from multiple LLMs and incorporating domain expert verification may offer a more robust and trustworthy solution than relying solely on a single model. We consider a more systematic exploration of the reliability and choice of different LLMs is an important direction for future work.
>
> >**[W3 and Q2] The choice of using PLIP and CONCH as VLMs.**
>
> Vision-language models (VLMs) have experienced explosive growth in the visual domain, however, CLIP is still widely regarded as a standard backbone for few-shot learning research [1][2]. In this work, we further explore the applicability of CLIP and its variant in pathology domain, PLIP, as our backbone. PLIP retains the architecture of CLIP while being fine-tuned on large-scale pathology-specific image-text data, making it suitable for investigating the effectiveness of prompt learning in few-shot WSI classification. Compared to CONCH, which is based on the more complex CoCa architecture, PLIP has lighter parameters and a structure more consistent with CLIP, facilitating direct and fair comparisons.
>
> We appreciate the reviewer’s suggestion to consider other strong pathology foundation models such as UNI, GPFM, and Virchow. However, we note that these are image-only models that do not include a compatible text encoder, and therefore cannot be directly integrated into our prompt-learning pipeline, which relies on vision–language alignment. To this end, we instead conduct the ablation study using pathology VLMs, including CLIP-based methods (QuiltNet, PLIP) and the CoCa-based method (CONCH). As shown in the table below, CONCH provides the best classification performance as it is a stronger foundation model.
>
> Next, for the purpose of illustrating the advantage of our MAPLE under different foundation models, we also compare it with all baselines using CONCH as the backbone. The experimental results shown in the response to [Q4] of Reviewer TGVt clearly verify the advantage of our MAPLE for few-shot WSI classification. We will include these CONCH-based results in the revised paper.
>
> In summary, we can observe that the choice of different VLM backbones could affect the few-shot WSI classification results. However, our MAPLE can achieve consistently superior prediction results under different vision-language foundation models (e.g., CLIP, PLIP, CONCH), highlighting the advantages of our MAPLE to jointly integrate multi-scale visual semantics and perform prediction at both the entity and slide levels.
>
> |Dataset|Methods|AUC(RCC)|F1(RCC)|ACC(RCC)|AUC(BRCA)|F1(BRCA)|ACC(BRCA)|AUC(NSCLC)|F1(NSCLC)|ACC(NSCLC)|
> |-|-|-|-|-|-|-|-|-|-|-|
> |**4-shot**|MAPLE-QuiltNet|0.688±0.063|0.560±0.050|0.630±0.068|0.914±0.017|0.720±0.060|0.748±0.061|0.728±0.063|0.641±0.086|0.651±0.064|
> ||MAPLE-PLIP|0.722±0.063|0.594±0.076|0.664±0.134|0.909±0.020|0.705±0.055|0.728±0.057|0.740±0.056|0.663±0.052|0.675±0.053|
> ||MAPLE-CONCH|0.844±0.109|0.653±0.126|0.695±0.166|0.947±0.017|0.791±0.086|0.805±0.069|0.889±0.055|0.774±0.032|0.786±0.033|
> |**8-shot**|MAPLE-QuiltNet|0.760±0.107|0.595±0.075|0.653±0.090|0.948±0.016|0.794±0.041|0.824±0.038|0.832±0.049|0.742±0.044|0.748±0.040|
> ||MAPLE-PLIP|0.786±0.070|0.618±0.024|0.673±0.018|0.957±0.015|0.791±0.024|0.806±0.024|0.855±0.041|0.762±0.031|0.766±0.030|
> ||MAPLE-CONCH|0.900±0.082|0.748±0.110|0.797±0.120|0.971±0.011|0.888±0.025|0.899±0.023|0.964±0.032|0.894±0.033|0.894±0.034|
> |**16-shot**|MAPLE-QuiltNet|0.791±0.027|0.660±0.033|0.715±0.033|0.964±0.009|0.846±0.032|0.884±0.027|0.891±0.039|0.805±0.052|0.806±0.052|
> ||MAPLE-PLIP|0.801±0.031|0.672±0.076|0.735±0.039|0.969±0.014|0.838±0.034|0.867±0.031|0.903±0.033|0.806±0.060|0.810±0.055|
> ||MAPLE-CONCH|0.916±0.024|0.790±0.029|0.825±0.030|0.984±0.006|0.910±0.016|0.919±0.015|0.981±0.005|0.914±0.050|0.914±0.047|
>
> [1] Zeng et al. Local-Prompt: Extensible Local Prompts for Few-Shot Out-of-Distribution Detection. ICLR 2025.\
> [2] Pan et al. NLPrompt: Noise-Label Prompt Learning for Vision-Language Models. CVPR 2025.
>
> >**[Q3] Results in Table 1 are relatively low, which stems from the choice of feature extractor.**
>
> We appreciate the reviewer’s observation. We agree that CONCH is a significantly stronger foundation model compared to PLIP, which may partly explain the AUC difference observed in Table 1 of original paper. To address this, we conduct additional experiments using CONCH as the visual encoder, with results presented in the response to [Q4] of Reviewer TGVt. Notably, MAPLE still achieves the best performance under this stronger backbone, reinforcing the effectiveness and generalizability of our method across different VLMs. We will include these CONCH-based results in the revised paper.
>
> >**[Q4] Explanation on visualization details in Section 4.3.**
>
> The visualized patches are selected based on attention scores from the entity-guided cross-attention module. For each entity, we rank the attention scores and visualize the top-k scoring patches, which reflect the most relevant regions associated with that entity. We will clarify this detail explicitly in Section 4.3 of the revised paper.
>
> >**[Q5] The point of using the global description and WSI-level representation if the entity can be precisely identified.**
>
> In MAPLE, entity-level prompts are designed to capture fine-grained and localized entities. However, WSIs are inherently heterogeneous, and accurate diagnosis often requires integrating evidence from multiple spatial and semantic cues, rather than relying on isolated entities alone. For instance, the presence of necrosis alone may not be diagnostically conclusive. However, when necrosis co-occurs with abnormal mitotic activity and specific inflammatory patterns, the combination strongly supports malignancy or tumor subtyping [1]. In such cases, global representations are essential for capturing diagnostic dependencies and interrelationships across the entire slide. Based on the above consideration, our slide-level representation is designed specifically to provide a holistic view that aggregates entity features. As confirmed in Table 2 of the main paper, MAPLE combined with entity-level and slide-level representations consistently outperforms using either alone, highlighting the effectiveness of both entity-level and slide-level information.
>
> [1] Wesseling et al. The pathological diagnosis of diffuse gliomas: towards a smart synthesis of microscopic and molecular information in a multidisciplinary context[J]. Diagnostic Histopathology, 2011, 17(11): 486-494.

---

> > ### Comment · Reviewer_yzxQ · 2025-08-01
> >
> > Thanks for the comprehensive response. The authors have addressed my concerns, and I'll raise the score.
> >
> > In the revised manuscript, please incorporate the aforementioned revision. Also, please move the Related Works part into the main text instead of the Appendix if possible.

---

> > > ### Author Response · Authors · 2025-08-01
> > >
> > > Thank you for considering our response and raising the score! We are glad to know that our rebuttal has addressed your concerns. We will incorporate the revisions and move the Related Works part into the main text in the revised paper.

---

### Official Review · Reviewer_RUWi · 2025-07-02

**Clarity:** 3
**Significance:** 2
**Originality:** 3
**Rating:** 5
**Confidence:** 4

**Summary:**

Few-shot WSI classifiers often miss fine-grained, subtype-specific histological variations crucial for accurate diagnosis. To remedy this, the authors introduce MAPLE, which uses a frozen language model to generate both entity-level prompts for detailed histological attributes and slide-level prompts for global tissue context. A cross-attention module guided by these prompts, together with a cross-scale entity graph, refines and aligns multi-resolution features before aggregating them for prediction, boosting both accuracy and interpretability.

**Questions:**

1. Prompt reliability seems crucial for this study. How did the authors generate prompts that prevent hallucinations or the omission of clinically critical attributes? It is helpful to provide results with and without expert correction to quantify the performance gains from text guidance. The authors also claim that MAPLE’s hierarchy mirrors diagnostic practice, yet they offer no user study or expert evaluation / references to validate this alignment.

2. Although qualitative examples show that selected patches visually match the LLM-described attributes, there are no region-level ground-truth annotations with heatmap visualizations (or IoU / Dice metrics) to demonstrate slide-level alignment.

3. Both the low-resolution-only and high-resolution-only models perform nearly as well as the multi-scale model, which only marginally outperforms the single-scale variants. How can we assess that each scale provides complementary information rather than redundant features? Furthermore, how can we ensure that 10× magnification captures sufficient cellular morphology detail compared with 20× or 40×?

4. The paper does not compare MAPLE against existing multi-scale MIL methods [1, 2, 3]. Could the authors include these baselines to isolate the benefits of their language-guided enhancements?

[1] DTFD-MIL: Double-Tier Feature Distillation Multiple Instance Learning for Histopathology Whole Slide Image Classification
[2] Dual-stream Multiple Instance Learning Network for Whole Slide Image Classification with Self-supervised Contrastive Learning
[3] Cross-scale multi-instance learning for pathological image diagnosis

**Ethical Concerns:**

["NO or VERY MINOR ethics concerns only"]

**Final Justification:**

The author addressed all my concerns with explicit illustrations and supporting experiments, which have improved the quality of the manuscript. Therefore, I would like to increase my score and recommend accepting the paper.

**Limitations:**

MAPLE depends on uncurated LLM prompts, risking hallucinations without expert oversight. It lacks pixel-level annotations for rigorous, quantitative region–prompt validation, relying instead on indirect ablation and qualitative examples.

**Quality:**

3

**Strengths And Weaknesses:**

1. Strengths:

(1) MAPLE consistently outperforms leading MIL and prompt-learning baselines across different few-shot settings on multiple datasets.

(2) By fusing entity-level features from both low and high magnification with slide-level cues, MAPLE surpasses single-scale variants. Ablation studies show that every key module (language-guided instance selection, entity-guided cross-attention, and cross-scale graph learning) contributes positively.

(3) With just less trainable parameters and inference/training times comparable to simpler baselines, MAPLE strikes a practical balance between accuracy and efficiency.

2. Weaknesses:

(1) Prompt quality depends entirely on raw LLM descriptions, which can hallucinate or omit clinically crucial details. The authors note this limitation and suggest future human-in-the-loop refinement.

(2) Without ground-truth entity labels, the paper lacks directly quantification on how well selected patches match textual prompts—relying instead on indirect ablation and qualitative figures.

(3) Although the paper highlights gains from multi-scale integration, it doesn’t benchmark against established multi-scale MIL architectures, leaving it unclear how much improvement stems from language cues versus simply using multiple resolutions.

---

> ### Author Rebuttal · Authors · 2025-07-31
>
> We thank the reviewer for the comments and for the time spent reviewing our paper. We address the weaknesses (W) and questions (Q) as follows:
>
> ---
>
> >**[W1 and Q1] Prompt reliability and MAPLE’s hierarchy.**
>
> We appreciate the reviewer’s insightful comments regarding the reliability of LLM-generated prompts and the clinical plausibility of MAPLE’s hierarchical design.
>
> Regarding the first point, we address this concern in detail in our response to [W3] of Reviewer dMRb. In current step, we carefully design the LLM instructions in the Prompt Construction stage and evaluate the reliability of GPT-4 outputs by querying multiple independent LLMs (e.g., Claude 3.5 Sonnet, DeepSeek-V3, and Qwen2.5), aiming to minimize the potential impact of hallucinations and increase the trustworthiness of the extracted entities. In future work, we plan to incorporate expert correction and clinical guidelines to further refine and validate entity construction.
>
> On the second point, MAPLE’s hierarchical design is motivated by real-world diagnostic workflows. We clarify this hierarchy and provide more explanations to validate it. Specifically, the hierarchy in MAPLE is reflected in two aspects:  multi-scale (high magnification and low resolution) and multi-level (entity-level and slide-level) design:
> * One the one hand, at high magnification, pathologists analyze cellular components such as nuclear pleomorphism and cytoplasmic features for cancer diagnosis. As to the image at low magnification, they distinguish different cancer subtypes by examining tissue architecture such as gland formation and tumor-stroma interfaces [1][2].
> * On the other hand, clinical diagnosis is rarely based on isolated observations. Instead, pathologists synthesize cues from multiple key tissue entities (e.g., mitotic activity, necrosis, glandular structures) and integrate them with global tissue context to reach a diagnosis [3]. For instance, necrosis alone may be inconclusive, but when co-occurring with abnormal mitoses and specific inflammatory patterns, the combination supports malignancy or tumor subtyping [4]. Our design emulates this diagnostic reasoning by jointly modeling localized entity-level and global slide-level descriptions.
>
> [1] Kumar et al. Whole slide imaging (wsi) in pathology: current perspectives and future directions. Journal of digital imaging, 33(4):1034–1040, 2020.\
> [2] Li et al. A multi-resolution model for histopathology image classification and localization with multiple instance learning. Computers in biology and medicine, 131:104253, 2021.\
> [3] Heba et al. A comprehensive review of the deep learning-based tumor analysis approaches in histopathological images: segmentation, classification and multi-learning tasks. Cluster Computing 2023.\
> [4] Wesseling et al. The pathological diagnosis of diffuse gliomas: towards a smart synthesis of microscopic and molecular information in a multidisciplinary context[J]. Diagnostic Histopathology, 2011, 17(11): 486-494.
>
> >**[W2 and Q2] Lack direct quantification on how well selected patches match textual prompts.**
>
> To the best of our knowledge, public WSI datasets typically lack spatial annotations that directly correspond to textual entities, making it infeasible to compute IoU or Dice metrics. In our current setup, we follow common practice adopted by prior works [1, 2], where semantic alignment is indirectly validated by visualizing the top-k patches with the highest attention scores from the entity-guided module and confirming that they visually reflect the corresponding textual descriptions.
>
> To further address the reviewer’s concern and provide quantitative insights, we additionally compute the similarity scores between selected patches and their corresponding entity descriptions. Specifically, for each entity, we select the top-50 and bottom-50 patches based on the similarity scores from entity-guided attention, and then use the CONCH model to compute similarity scores with the entity description and the patches from different sets. Finally, we can derive the average scores of the top-50 and bottom-50 patches. We report the scores for the examples (Figure 4 of the original paper) in the table below. The results confirm that the selected patches align well with their associated textual descriptions, further validating the effectiveness of our method.
>
> ||Stroma (TCGA-55-6970 from LUAD)|Nucleolus (TCGA-55-6970 from LUAD)|Stroma  (TCGA-58-A46M from LUSC)|Nucleolus  (TCGA-58-A46M from LUSC)|
> |-|-|-|-|-|
> |top-50 score|0.32|0.34|0.25|0.27|
> |bottom-50 score|0.15|0.12|0.08|0.07|
>
> [1] Lu et al. Visual Language Pretrained Multiple Instance Zero-Shot Transfer for Histopathology Images. CVPR 2023.\
> [2] Jaume et al. Modeling Dense Multimodal Interactions Between Biological Pathways and Histology for Survival Prediction. CVPR 2024.
>
> >**[Q3] Complementarity of multi-scale features and selection of magnifications.**
>
> MAPLE is designed to capture distinct histological entities at different magnifications. Different resolution levels in WSIs naturally correspond to different semantic scales: low magnification (e.g., 5×) reveals tissue architecture and global organizational patterns, while high magnification (e.g., 10×) focuses on cellular morphology and nuclear detail. These features are inherently complementary rather than redundant. As demonstrated in Table 2 of the original paper, integrating both scales consistently outperforms using either scale alone. To further validate this observation, we perform the paired t-test comparing multi-scale and single-scale results. The resulting p-values are consistently below 0.05, confirming that the performance gains from multi-scale integration are statistically significant.
>
> Our resolution choices follow ViLa-MIL [1], which also employs dual-scale inputs at 5× and 10× for few-shot slide-level classification. We agree that higher magnification (e.g., 20×) may offer finer morphological cues. To explore this, we conduct experiments using 20× as the high resolution, and find that 10× and 20× perform comparably, suggesting that both magnifications can effectively capture high-resolution histological features. Considering that 20× images significantly increase memory usage and computational cost, we adopt 10× in our main experiments to achieve a balance between performance and efficiency.
>
> |Dataset|Methods|AUC(RCC)|F1(RCC)|ACC(RCC)|AUC(BRCA)|F1(BRCA)|ACC(BRCA)|AUC(NSCLC)|F1(NSCLC)|ACC(NSCLC)|
> |-|-|-|-|-|-|-|-|-|-|-|
> |**4-shot**|MAPLE-20×|0.714±0.052|0.583±0.118|0.647±0.181|0.904±0.027|0.691±0.075|0.717±0.080|0.739±0.061|0.654±0.121|0.666±0.074|
> ||MAPLE-10×|0.722±0.063|0.594±0.076|0.664±0.134|0.909±0.020|0.705±0.055|0.728±0.057|0.740±0.056|0.663±0.052|0.675±0.053|
> |**8-shot**|MAPLE-20x|0.784±0.075|0.621±0.092|0.680±0.109|0.958±0.013|0.796±0.060|0.813±0.062|0.847±0.073|0.751±0.088|0.760±0.082|
> ||MAPLE-10×|0.786±0.070|0.618±0.024|0.673±0.018|0.957±0.015|0.791±0.024|0.806±0.024|0.855±0.041|0.762±0.031|0.766±0.030|
> |**16-shot**|MAPLE-20x|0.796±0.034|0.688±0.077|0.746±0.098|0.960±0.012|0.831±0.019|0.857±0.021|0.909±0.041|0.793±0.022|0.805±0.023|
> ||MAPLE-10×|0.801±0.031|0.672±0.076|0.735±0.039|0.969±0.014|0.838±0.034|0.867±0.031|0.903±0.033|0.806±0.060|0.810±0.055|
>
> [1] Shi et al. Vila-mil: Dual-scale vision-language multiple instance learning for whole slide image classification. CVPR 2024.
>
> >**[W3 and Q4] Comparison against existing multi-scale MIL methods.**
>
> We appreciate the reviewer’s suggestion and conduct additional experiments to include existing multi-scale MIL baselines such as DTFD-MIL, Dual-Stream MIL, and Cross-Scale MIL under the same 5×/10× magnification settings used in MAPLE. As shown in the table below, MAPLE consistently outperforms these baselines across all datasets, highlighting the advantage of our multi-scale prompt-guided VLM-based method. These results further confirm that the performance gain of MAPLE arises not merely from multi-scale integration, but from the combination of multi-scale modeling and language-guide prompt supervision via vision-language models.
>
> |Dataset|Methods|AUC(RCC)|F1(RCC)|ACC(RCC)|AUC(BRCA)|F1(BRCA)|ACC(BRCA)|AUC(NSCLC)|F1(NSCLC)|ACC(NSCLC)|
> |-|-|-|-|-|-|-|-|-|-|-|
> |**4-shot**|DTFD-MIL|0.648±0.050|0.520±0.065|0.593±0.068|0.869±0.031|0.626±0.057|0.660±0.070|0.624±0.049|0.555±0.051|0.581±0.046|
> ||Dual-Stream MIL|0.669±0.052|0.562±0.132|0.623±0.162|0.877±0.039|0.664±0.037|0.693±0.036|0.649±0.122|0.564±0.094|0.590±0.080|
> ||Cross-Scale MIL|0.673±0.097|0.552±0.061|0.615±0.068|0.876±0.019|0.667±0.037|0.690±0.039|0.651±0.103|0.560±0.093|0.586±0.084|
> ||MAPLE|0.722±0.063|0.594±0.076|0.664±0.134|0.909±0.020|0.705±0.055|0.728±0.057|0.740±0.056|0.663±0.052|0.675±0.053|
> |**8-shot**|DTFD-MIL|0.733±0.048|0.546±0.061|0.603±0.086|0.907±0.024|0.722±0.036|0.748±0.046|0.729±0.040|0.632±0.042|0.652±0.030|
> ||Dual-Stream MIL|0.758±0.073|0.548±0.071|0.576±0.088|0.926±0.025|0.761±0.041|0.789±0.037|0.752±0.074|0.651±0.032|0.667±0.034|
> ||Cross-Scale MIL|0.756±0.062|0.554±0.063|0.588±0.075|0.924±0.023|0.757±0.028|0.782±0.031|0.748±0.037|0.645±0.075|0.657±0.071|
> ||MAPLE|0.786±0.070|0.618±0.024|0.673±0.018|0.957±0.015|0.791±0.024|0.806±0.024|0.855±0.041|0.762±0.031|0.766±0.030|
> |**16-shot**|DTFD-MIL|0.738±0.044|0.623±0.064|0.679±0.088|0.919±0.019|0.762±0.051|0.799±0.040|0.812±0.047|0.742±0.044|0.747±0.042|
> ||Dual-Stream MIL|0.752±0.038|0.636±0.059|0.696±0.062|0.946±0.014|0.813±0.030|0.827±0.032|0.824±0.029|0.760±0.037|0.762±0.032|
> ||Cross-Scale MIL|0.759±0.052|0.632±0.055|0.698±0.060|0.948±0.019|0.815±0.034|0.830±0.035|0.830±0.042|0.764±0.029|0.768±0.028|
> ||MAPLE|0.801±0.031|0.672±0.076|0.735±0.039|0.969±0.014|0.838±0.034|0.867±0.031|0.903±0.033|0.806±0.060|0.810±0.055|

---

> > ### Comment · Reviewer_RUWi · 2025-08-07
> >
> > Thank you so much for the author’s efforts in providing explicit illustrations and additional experiments to address my concerns. Please make sure to include all of the clarifications and new experiments in the final version of the manuscript. I would like to increase my score to reflect the improvements in the quality of the manuscript.

---

> > > ### Author Response · Authors · 2025-08-07
> > >
> > > We sincerely thank the reviewer for the positive feedback and for acknowledging our efforts. We are pleased to hear that the revisions have addressed your concerns. We will ensure that all clarifications and new experimental results are included in the final version. We greatly appreciate your thoughtful review and the revised score.

---

### Official Review · Reviewer_dMRb · 2025-07-04

**Clarity:** 4
**Significance:** 4
**Originality:** 3
**Rating:** 5
**Confidence:** 4

**Summary:**

This paper addresses the limitation of existing few-shot WSI classification methods that lack fine-grained, entity-level prompts. It proposes a Multi-scale Attribute-enhanced Prompt Learning approach, which incorporates both slide-level and entity-level prompts. The method fully leverages the capabilities of large language models (LLMs) throughout the process. Furthermore, an entity graph is constructed to enhance instance features and derive entity-level representations. Finally, the predictions from both the entity-level and slide-level are integrated to produce the final classification result.

**Questions:**

please refer to Weaknesses

**Ethical Concerns:**

["NO or VERY MINOR ethics concerns only"]

**Final Justification:**

The authors’ responses have addressed most of my concerns. I will maintain my score.

**Limitations:**

please refer to Weaknesses

**Quality:**

3

**Strengths And Weaknesses:**

Strengths:
1. To achieve prompt learning aligned with fine-grained features, this paper designs components such as entity name extraction, hierarchical entity graphs, and an entity-guided cross-attention module—all of which are validated through ablation studies.
 2. By introducing entity-level prompt learning, the paper enables more interpretable predictions at the entity level.

Weaknesses:
1. The font size in Figure 1 is too small and difficult to read; it is recommended to enlarge the text for better readability.
2. In Figure 2, the meaning of the arrows between the entity-level features and the learnable prompts is unclear. It is suggested to add symbols or annotations to clarify their purpose.
3. How does the use of LLM ensure comprehensive and accurate extraction of entity names? Is there a possibility of incomplete or missing entity names? If the extraction is not sufficiently comprehensive, will it affect model performance, and how is this issue addressed?
4. In the Region Selection module, is there a high degree of overlap among the selected instances, meaning that most image patches come from the same region? Is this phenomenon beneficial or detrimental to the final model performance?

---

> ### Author Rebuttal · Authors · 2025-07-31
>
> We thank the reviewer for the comments and for the time spent reviewing our paper. We address the weaknesses (W) as follows:
>
> ---
>
> >**[W1] The font size in Figure 1 is too small and difficult to read.**
>
> We thank the reviewer for pointing this out. We will revise Figure 1 by enlarging the font size and optimizing the layout to ensure clarity and readability in the final manuscript.
>
> >**[W2] In Figure 2, the meaning of the arrows between the entity-level features and the learnable prompts is unclear.**
>
> We appreciate the reviewer’s suggestion. The arrows represent the computation of similarity scores between the extracted entity-level features and the learnable prompts. We will revise Figure 2 by adding appropriate annotations to clarify their meaning and ensure interpretability.
>
> >**[W3] Reliability and completeness of entity extraction by LLMs.**
>
> We thank the reviewer for this insightful comment. To ensure comprehensive and accurate extraction of entities, we carefully design the LLM instructions in the Prompt Construction stage to prioritize entities with higher clinical importance. For example, we incorporate prompts such as "iteratively suggest a discriminative histological entity not in the current entity set" (see Appendix B.1 for details), thereby reducing the likelihood of missing critical entities. This prioritization encourages the LLM to focus on highly discriminative entities.
>
> To further evaluate the accuracy of the selected entities, we examine the entities generated from GPT4 by querying different LLMs (e.g., Claude 3.5 Sonnet, DeepSeek-V3 and Qwen2.5) using the prompt: "Do you think [ENTITY] is a key entity relevant to distinguishing different subtypes in [DATASET]?" All of these LLMs affirm the relevance and importance of the generated entities, which can demonstrate the accuracy of selected entities.
>
> Furthermore, as shown in our ablation study on the number of entities (Figure 8 in the original paper), increasing the number of entities beyond a certain threshold does not improve performance and may even slightly decrease. This suggests that redundant entities may lack discriminative phenotypic attributes relevant to cancer subtype classification and potentially introduce noise that hinders the model's decision-making process. This observation supports the completeness and effectiveness of our selected entity set.
>
> As acknowledged in our Limitations section, we recognize that LLM-generated entities may still suffer from hallucinations or generate clinically irrelevant descriptions in the absence of expert curation. To mitigate this, we propose the following future directions:
> - Incorporating expert review or clinical guidelines to refine and validate the extracted entities;
> - Aggregating outputs from multiple LLMs (e.g., a mixture-of-experts approach) to achieve more robust and reliable entity sets and descriptions rather than relying solely on a single model.
>
> We believe these strategies will enhance the reliability and clinical applicability of LLM-driven entity construction, and we plan to explore them in future versions of MAPLE.
>
> >**[W4] Patch overlap and the region selection module.**
>
> The Region Selection module in MAPLE is designed to accurately identify tumor-relevant regions while suppressing irrelevant background areas, thereby guiding the model to focus on informative patches that align well with the entity-level prompts. Since patch instances are partitioned in a non-overlapping manner and the region selection ratio is set to r = 0.7, the selected patches can effectively cover a variety of heterogeneous tumor-related regions within a WSI, and thus do not exhibit a high degree of spatial redundancy or overlap. We also investigate the impact of r in our ablation study (Section E.3 of the main paper). As shown in Figure 9, higher values of r (e.g., r > 0.7) yields the slight performance degradation. This decline can be attributed to the inclusion of less informative or non-tumor regions that compromise the quality of the selected patch set. In contrast, lower values (e.g., r < 0.7) tend to overly concentrate on a few localized regions, resulting in performance degradation.

---

> > ### Comment · Reviewer_dMRb · 2025-08-06
> >
> > The authors’ responses have addressed most of my concerns. I will maintain my score.

---

> > > ### Author Response · Authors · 2025-08-06
> > >
> > > Thank you for taking the time to review our responses. We sincerely appreciate your thoughtful feedback and your engagement throughout the review process.
> > >
> > > We are happy to address any further questions you may have!

---

### Official Review · Reviewer_TGVt · 2025-07-08

**Clarity:** 4
**Significance:** 3
**Originality:** 3
**Rating:** 4
**Confidence:** 3

**Summary:**

The work presents prompting-based learning for few-shot WSI classification. An LLM is used to form prompts at both the entity level and the slide level, at both low and high resolutions. A cross-attention mechanism is used to extract features for each entity, and cosine similarity is then computed with subtype-specific prompts to generate logits for classification. A Graph Attention Network is employed to account for correlations among various entities. Additionally, slide-level representation is formed in a similar manner, where the LLM is prompted to identify slide-level features. Slide-level logits and entity-level logits are then averaged using a parameter lambda to produce the final logits. The effectiveness of the proposed method is evaluated on three WSI datasets across three few-shot settings.

**Questions:**

1) Please include baseline results for ABMIL and TransMIL using state-of-the-art pathology foundation models such as Virchow 2 or UNI2-h in Table 1, evaluated across the three datasets and all three few-shot settings. The PLIP model used in this study is a relatively much weaker feature extractor.

2) Provide results for the 32-shot, 64-shot, and fully supervised (all-label) settings, and compare them with the existing baselines in Table 1. These results will offer a clearer perspective on the generalizability of the proposed method across varying levels of label availability.

3) Include a comparison of the proposed method with ConcepPath across multiple few-shot settings (Table 1) to highlight differences in performance.

4) Provide an ablation study evaluating the effect of using different VLMs, such as CONCH or QuiltNet, on the performance of both the proposed method and baseline methods, at least on one representative dataset.

**Ethical Concerns:**

["NO or VERY MINOR ethics concerns only"]

**Final Justification:**

I thank the authors for the thorough rebuttal.

**Limitations:**

yes

**Quality:**

4

**Strengths And Weaknesses:**

Strengths:

1) The method is clearly and effectively presented.

2) MAPLE explores the use of scale-specific prompts, which is intuitive and well-motivated.

3) The integration of the Graph Attention Network (GAT) is both interesting and effective.

4) Exhaustive ablation studies are provided for each component of the method.


Weaknesses:

1) While the proposed method depends on a VLM-based pathology foundation model, the baselines using traditional MIL approaches such as ABMIL and TransMIL could benefit from stronger image-only foundation models like Virchow 2 or UNI2-h. The study lacks a fair comparison in which MIL methods are trained using these state-of-the-art pathology feature encoders.

2) All experiments are conducted with at most a 16-shot setting, which limits the generalizability of the proposed approach to more realistic scenarios where around fifty to several hundred WSIs per class may be available.

3) The paper lacks any comparison with the ConcepPath method, which is highly relevant as it also decomposes slide-level prompts into concept-level prompts and proposes a guided aggregation strategy. Although ConcepPath was developed under a fully supervised setting, it can be readily adapted to few-shot scenarios as well.

---

> ### Author Rebuttal · Authors · 2025-07-31
>
> We thank the reviewer for the comments and for the time spent reviewing our paper. We address the weaknesses (W) and questions (Q) as follows:
>
> ---
>
> >**[W1 and Q1] SOTA pathology foundation models for MIL baselines.**
>
> We agree that traditional MIL approaches like ABMIL and TransMIL could indeed benefit from stronger image-only foundation models such as Virchow 2 or UNI2-h. In our main experiments (Table 1 and 5), we use the same feature extractor (i.e., PLIP and CLIP) across all baselines and MAPLE to ensure a fair comparison. To address the reviewer's concern, we additionally conduct the experiment including ABMIL and TransMIL with stronger image-only foundation models (e.g., UNI2-h), and MAPLE with a stronger VLM (e.g., CONCH). As shown in the table below, we can derive the following observations. Firstly, both ABMIL and TransMIL with UNI2-h exhibit noticeable performance gains, validating the importance of encoder strength. Secondly, MAPLE with CONCH can consistently beat the MIL based methods with latest image-only foundation model (i.e., UNI2-h), demonstrating the scalability and effectiveness of our method. Moreover, we compare MAPLE with all baseline methods under CONCH for a fair comparison in the response to [Q4]. MAPLE achieves the best performance across different few-shot settings and datasets, highlighting its effectiveness under different VLMs. In summary, these findings confirm that while backbone strength contributes to overall performance, MAPLE consistently delivers superior results.
> ||Methods|AUC(RCC)|F1(RCC)|ACC(RCC)|AUC(BRCA)|F1(BRCA)|ACC(BRCA)|AUC(NSCLC)|F1(NSCLC)|ACC(NSCLC)|
> |-|-|-|-|-|-|-|-|-|-|-|
> |**4-shot**|ABMIL-UNI2|0.787±0.045|0.604±0.073|0.651±0.085|0.932±0.012|0.763±0.031|0.781±0.031|0.806±0.054|0.704±0.029|0.713±0.028|
> ||TransMIL-UNI2|0.779±0.075|0.587±0.198|0.631±0.228|0.941±0.017|0.768±0.046|0.785±0.058|0.815±0.078|0.719±0.089|0.729±0.081|
> ||MAPLE-CONCH|0.844±0.109|0.653±0.126|0.695±0.166|0.947±0.017|0.791±0.086|0.805±0.069|0.889±0.055|0.774±0.032|0.786±0.033|
> |**8-shot**|ABMIL-UNI2|0.855±0.030|0.710±0.068|0.742±0.070|0.953±0.008|0.861±0.024|0.872±0.020|0.927±0.023|0.819±0.038|0.830±0.038|
> ||TransMIL-UNI2|0.847±0.052|0.701±0.062|0.730±0.073|0.961±0.002|0.870±0.011|0.878±0.014|0.915±0.045|0.808±0.068|0.811±0.065|
> ||MAPLE-CONCH|0.900±0.082|0.748±0.110|0.797±0.120|0.971±0.011|0.888±0.025|0.899±0.023|0.964±0.032|0.894±0.033|0.894±0.034|
> |**16-shot**|ABMIL-UNI2|0.884±0.031|0.767±0.048|0.790±0.040|0.962±0.003|0.865±0.012|0.868±0.005|0.961±0.011|0.891±0.023|0.897±0.023|
> ||TransMIL-UNI2|0.897±0.026|0.772±0.104|0.804±0.121|0.969±0.002|0.874±0.011|0.879±0.012|0.949±0.026|0.885±0.036|0.885±0.036|
> ||MAPLE-CONCH|0.916±0.024|0.790±0.029|0.825±0.030|0.984±0.006|0.910±0.016|0.919±0.015|0.981±0.005|0.914±0.050|0.914±0.047|
>
> >**[Q2 and W2] Generalizability beyond 16-shot settings.**
>
> For few-shot WSI classification, we follow prior works (e.g., MSCPT and FOCUS) and set the shots as 4, 8 and 16. We agree that evaluating the performance under larger-shot settings (e.g., 32-shot and 64-shot) is helpful for assessing MAPLE to more realistic scenarios where more WSIs per class are available. To this end, we conduct additional experiments under 32-shot and 64-shot settings, and present the results in the table below. As shown, MAPLE continues to outperform existing baselines, demonstrating its generalizability under larger-shot settings. Due to character limitation, we report the fully supervised results directly in the text. Specifically, our MAPLE achieves the best AUC of 0.903, 0.990 and 0.966 across three datasets, demonstrating significant improvements over the second-best method FOCUS with AUC of 0.880, 0.975 and 0.955.
> ||Methods|AUC(RCC)|F1(RCC)|ACC(RCC)|AUC(BRCA)|F1(BRCA)|ACC(BRCA)|AUC(NSCLC)|F1(NSCLC)|ACC(NSCLC)|
> |-|-|-|-|-|-|-|-|-|-|-|
> |**32-shot**|ViLa-MIL|0.830±0.029|0.704±0.018|0.775±0.023|0.954±0.014|0.806±0.020|0.837±0.023|0.892±0.017|0.809±0.018|0.809±0.018|
> ||MSCPT|0.816±0.038|0.670±0.044|0.729±0.052|0.949±0.014|0.801±0.054|0.828±0.051|0.899±0.028|0.823±0.036|0.823±0.036|
> ||FOCUS|0.826±0.021|0.696±0.032|0.769±0.050|0.961±0.009|0.836±0.037|0.849±0.032|0.909±0.028|0.850±0.028|0.851±0.028|
> ||MAPLE|0.849±0.025|0.727±0.050|0.784±0.065|0.975±0.015|0.855±0.049|0.869±0.038|0.926±0.022|0.856±0.029|0.857±0.029|
> |**64-shot**|ViLa-MIL|0.859±0.025|0.715±0.044|0.772±0.048|0.966±0.003|0.861±0.012|0.884±0.012|0.924±0.006|0.851±0.010|0.851±0.010|
> ||MSCPT|0.849±0.014|0.687±0.036|0.742±0.040|0.953±0.005|0.857±0.018|0.864±0.013|0.928±0.013|0.853±0.021|0.854±0.021|
> ||FOCUS|0.852±0.017|0.710±0.035|0.772±0.036|0.967±0.004|0.864±0.014|0.885±0.013|0.930±0.018|0.859±0.015|0.861±0.014|
> ||MAPLE|0.873±0.025|0.733±0.027|0.796±0.034|0.980±0.001|0.889±0.018|0.899±0.014|0.942±0.031|0.876±0.029|0.876±0.029|
>
> >**[W3 and Q3] Comparison with ConcepPath.**
>
> We thank the reviewer for pointing out the relevance of ConcepPath. We report the results of ConcepPath in the table below. MAPLE consistently outperforms ConcepPath across different datasets and few-shot settings, highlighting the effectiveness of our method in few-shot WSI classification.
> ||Methods|AUC(RCC)|F1(RCC)|ACC(RCC)|AUC(BRCA)|F1(BRCA)|ACC(BRCA)|AUC(NSCLC)|F1(NSCLC)|ACC(NSCLC)|
> |-|-|-|-|-|-|-|-|-|-|-|
> |**4-shot**|ConcepPath|0.672±0.031|0.555±0.091|0.605±0.086|0.877±0.026|0.654±0.057|0.676±0.043|0.657±0.045|0.601±0.053|0.604±0.031|
> ||MAPLE|0.722±0.063|0.594±0.076|0.664±0.134|0.909±0.020|0.705±0.055|0.728±0.057|0.740±0.056|0.663±0.052|0.675±0.053|
> |**8-shot**|ConcepPath|0.759±0.072|0.571±0.093|0.635±0.142|0.920±0.032|0.753±0.068|0.774±0.066|0.750±0.034|0.668±0.033|0.680±0.035|
> ||MAPLE|0.786±0.070|0.618±0.024|0.673±0.018|0.957±0.015|0.791±0.024|0.806±0.024|0.855±0.041|0.762±0.031|0.766±0.030|
> |**16-shot**|ConcepPath|0.764±0.068|0.631±0.073|0.694±0.069|0.938±0.031|0.817±0.035|0.835±0.032|0.831±0.053|0.753±0.046|0.756±0.039|
> ||MAPLE|0.801±0.031|0.672±0.076|0.735±0.039|0.969±0.014|0.838±0.034|0.867±0.031|0.903±0.033|0.806±0.060|0.810±0.055|
>
> >**[Q4] Evaluating the effect of using different VLMs.**
>
> In addition to the experiments using CLIP and PLIP in the original paper, we provide further results with stronger pathology VLMs such as CONCH in the table below. MAPLE consistently outperforms baseline methods across different datasets and few-shot settings, highlighting the effectiveness of our method. We will include these results in the revised paper.
> Furthermore, we conduct a comprehensive ablation study evaluating the impact of different pathology VLMs including QuiltNet, PLIP, and CONCH, as detailed in our response to [W3 and Q2] of Reviewer yzxQ.
> ||Methods|AUC(RCC)|F1(RCC)|ACC(RCC)|AUC(BRCA)|F1(BRCA)|ACC(BRCA)|AUC(NSCLC)|F1(NSCLC)|ACC(NSCLC)|
> |-|-|-|-|-|-|-|-|-|-|-|
> |**4-shot**|ABMIL|0.770±0.069|0.591±0.070|0.648±0.105|0.924±0.026|0.755±0.063|0.773±0.062|0.832±0.041|0.714±0.049|0.721±0.044|
> ||TransMIL|0.757±0.127|0.602±0.111|0.672±0.126|0.938±0.019|0.761±0.098|0.778±0.099|0.848±0.059|0.756±0.125|0.762±0.109|
> ||GTMIL|0.728±0.102|0.552±0.094|0.630±0.141|0.923±0.031|0.743±0.070|0.765±0.054|0.836±0.093|0.750±0.080|0.754±0.081|
> ||WiKG|0.768±0.109|0.605±0.105|0.651±0.123|0.925±0.010|0.761±0.038|0.777±0.036|0.820±0.087|0.730±0.084|0.733±0.083|
> ||TOP|0.728±0.170|0.583±0.117|0.638±0.105|0.916±0.033|0.743±0.057|0.758±0.053|0.816±0.066|0.683±0.130|0.707±0.092|
> ||ViLa-MIL|0.783±0.108|0.590±0.110|0.635±0.130|0.919±0.030|0.768±0.058|0.793±0.051|0.853±0.073|0.759±0.081|0.756±0.082|
> ||MSCPT|0.782±0.087|0.605±0.072|0.632±0.088|0.931±0.019|0.770±0.057|0.785±0.050|0.842±0.059|0.730±0.068|0.748±0.066|
> ||FOCUS|0.810±0.115|0.632±0.135|0.667±0.160|0.930±0.032|0.767±0.075|0.780±0.065|0.875±0.077|0.762±0.060|0.769±0.061|
> ||MAPLE|0.844±0.109|0.653±0.126|0.695±0.166|0.947±0.017|0.791±0.086|0.805±0.069|0.889±0.055|0.774±0.032|0.786±0.033|
> |**8-shot**|ABMIL|0.857±0.049|0.705±0.057|0.763±0.058|0.941±0.013|0.835±0.055|0.844±0.047|0.925±0.013|0.835±0.021|0.835±0.021|
> ||TransMIL|0.853±0.044|0.710±0.044|0.771±0.046|0.940±0.012|0.840±0.030|0.851±0.024|0.916±0.010|0.821±0.027|0.821±0.027|
> ||GTMIL|0.861±0.052|0.711±0.068|0.780±0.065|0.945±0.006|0.845±0.020|0.857±0.016|0.928±0.016|0.844±0.020|0.844±0.020|
> ||WiKG|0.851±0.045|0.685±0.050|0.741±0.060|0.947±0.012|0.834±0.025|0.855±0.020|0.919±0.007|0.834±0.021|0.834±0.020|
> ||TOP|0.859±0.028|0.701±0.050|0.758±0.054|0.929±0.020|0.814±0.039|0.828±0.035|0.908±0.043|0.817±0.068|0.818±0.067|
> ||ViLa-MIL|0.880±0.081|0.726±0.117|0.776±0.090|0.945±0.008|0.835±0.037|0.858±0.032|0.934±0.037|0.856±0.051|0.857±0.051|
> ||MSCPT|0.882±0.091|0.720±0.132|0.774±0.136|0.950±0.011|0.849±0.043|0.851±0.041|0.924±0.043|0.849±0.055|0.849±0.055|
> ||FOCUS|0.875±0.060|0.719±0.114|0.747±0.145|0.959±0.008|0.871±0.033|0.875±0.031|0.949±0.030|0.873±0.043|0.873±0.042|
> ||MAPLE|0.900±0.082|0.748±0.110|0.797±0.120|0.971±0.011|0.888±0.025|0.899±0.023|0.964±0.032|0.894±0.033|0.894±0.034|
> |**16-shot**|ABMIL|0.876±0.017|0.759±0.020|0.789±0.018|0.954±0.004|0.857±0.009|0.859±0.007|0.935±0.013|0.865±0.021|0.865±0.021|
> ||TransMIL|0.884±0.030|0.761±0.057|0.795±0.057|0.955±0.003|0.854±0.010|0.860±0.013|0.926±0.014|0.851±0.027|0.851±0.027|
> ||GTMIL|0.891±0.025|0.770±0.073|0.803±0.083|0.962±0.004|0.865±0.008|0.878±0.013|0.938±0.016|0.874±0.020|0.874±0.020|
> ||WiKG|0.882±0.013|0.762±0.019|0.796±0.018|0.957±0.010|0.839±0.031|0.856±0.031|0.939±0.007|0.864±0.021|0.864±0.020|
> ||TOP|0.887±0.011|0.768±0.016|0.790±0.015|0.944±0.003|0.829±0.027|0.835±0.027|0.924±0.013|0.859±0.025|0.859±0.025|
> ||ViLa-MIL|0.902±0.033|0.775±0.038|0.812±0.042|0.966±0.006|0.862±0.024|0.872±0.016|0.941±0.023|0.877±0.028|0.877±0.017|
> ||MSCPT|0.894±0.018|0.767±0.027|0.808±0.024|0.958±0.004|0.859±0.021|0.871±0.025|0.934±0.017|0.866±0.031|0.867±0.031|
> ||FOCUS|0.893±0.017|0.764±0.041|0.805±0.042|0.974±0.006|0.884±0.045|0.891±0.051|0.964±0.007|0.894±0.052|0.895±0.050|
> ||MAPLE|0.916±0.024|0.790±0.029|0.825±0.030|0.984±0.006|0.910±0.016|0.919±0.015|0.981±0.005|0.914±0.050|0.914±0.047|

---

> > ### Comment · Reviewer_TGVt · 2025-08-03
> >
> > I thank the authors for the thorough rebuttal. My concerns are addressed.

---

> > > ### Author Response · Authors · 2025-08-03
> > >
> > > Thank you for carefully reading our response and for your prompt feedback. We truly appreciate your thoughtful engagement with our work. If you have any further questions or concerns during your review, please don't hesitate to reach out - we're more than happy to provide additional clarification or information to assist in your assessment.

---

### Note · Authors · 2025-08-12

We sincerely thank all reviewers for their time, effort, and constructive feedback on our work. We greatly appreciate the recognition of our method's novelty, clarity, and effectiveness, as well as the thoughtful suggestions that have helped us further strengthen the manuscript. During the rebuttal process, we conducted additional experiments and analyses to address the raised concerns, and we are encouraged that these efforts have been acknowledged by the reviewers. These results, extended comparisons, and clarifications will be incorporated into the final version to further improve the quality of the manuscript. We once again thank the reviewers for their constructive engagement, which has been invaluable in enhancing this work.

---

### Decision · Program_Chairs · 2025-09-17

**Decision:**

Accept (poster)

**Comment:**

This paper proposes MAPLE, a multi-scale attribute-enhanced prompt learning framework for few-shot WSI classification. By combining entity-level prompts (generated via LLMs) with slide-level prompts, and refining features through entity-guided cross-attention and cross-scale graph learning, MAPLE achieves consistent improvements across multiple cancer datasets and few-shot regimes.

The paper is well-motivated and supported by extensive ablations. Reviewers initially raised concerns about fairness of baselines (stronger encoders, larger-shot settings, comparisons with ConcepPath and multi-scale MIL methods), clarity of figures, and reliability of LLM-generated prompts. The authors provided comprehensive additional experiments and clarifications during rebuttal, which convincingly addressed these points. While prompt reliability and lack of direct region–prompt alignment remain limitations, they are acknowledged with clear future directions.

Overall, reviewers converged towards acceptance after rebuttal, with multiple raising scores. I recommend accept as a solid and meaningful contribution, though not at spotlight/oral level given the prompt reliability and limited clinical validation.